# Effects of global versus local trunk muscle strength training on muscle strength, proxies of power and rowing-specific performance in pubertal male rowers

Raouf Hammami[1,2], Yassine Negra[1,3], Abdelkader Mahmoudi[1,2], Walid Selmi[1,3], Haithem Rebai[2], David G. Behm[4], Anis Chaouachi[1,2,5], Urs Granacher[6]*

1 University of Manouba, Higher Institute of Sport and Physical Education of Ksar-Said, University Campus, Manouba, Tunis, Tunisia, 2 Tunisian Research Laboratory 'Sports Performance Optimization' (CNMSS-LR09SEP01), National Center of Medicine and Science in Sports (CNMSS), Tunis, Tunisia, 3 Research Laboratory (LR23JS01) «Sport Performance, Health & Society», Higher Institute of Sport and Physical Education of Ksar-Said, Manouba University, Tunis, Tunisia, 4 Memorial University St John's, Newfoundland, Canada, 5 Sports Performance Research Institute New Zealand, Auckland University of Technology, Auckland, New Zealand, 6 Department of Sport and Sport Science, University of Freiburg, Exercise and Human Movement Science, Freiburg, Germany

* urs.granacher@sport.uni-freiburg.de

## Abstract

Strength training is fundamental during long-term athlete development to enhance strength and power supporting sport-specific performance. In rowing, trunk muscles stabilize the body and transmit forces between the lower and upper limbs. This study compared the effects of pre-season global (GST) versus local (LST) trunk strength training on muscle strength, power, and rowing-specific performance in young male rowers. Twenty-eight Tier 2 athletes aged 12–13 years (circa-PHV = 0.2–0.3) completed a 6-week program with two weekly sessions. GST involved machine-based and free-weight trunk exercises at 70% 1-RM, whereas LST emphasized body-weight trunk exercises on stable and unstable surfaces. Pre- and post-tests included lower- and upper-limb power, trunk strength, and a 700-m rowing ergometer test. Significant group-by-time interactions were found for all strength (d = 3.04–3.84; p < 0.001), power (d = 0.75–2.34; p < 0.01), and rowing performance outcomes (d = 1.61; p < 0.001). Post-hoc analyses indicated greater improvements in GST (d = 0.28–1.87; p < 0.001) than in LST (d = 0.11–0.73; p < 0.001). In conclusion, GST produced larger performance gains than LST. However, these effects likely reflect the combined influence of exercise modality and higher external loading intensity, rather than trunk muscle recruitment patterns alone. These findings should be interpreted with caution given that differences in external loading between conditions (70% 1-RM vs. athletes' body mass) confound the comparison of GST versus LST modalities. Strength and conditioning specialists may consider incorporating GST to enhance foundational strength and power in pubertal male rowers, but further research controlling for training load is needed to isolate the effects of exercise modality.

**Data availability statement:** All relevant data are within the paper and its Supporting Information files.

**Funding:** The author(s) received no specific funding for this work.

**Competing interests:** The authors have declared that no competing interests exist.

## Introduction

Strength training (ST) is a widely applied and effective method during long-term athlete development (LTAD) to build a foundation of muscle strength and power for subsequent sport-specific performance [1,2]. According to Haff [2], periodization strategies for young athletes in the early LTAD stages up to the "training to train" stage should emphasize general or foundational training rather than sport-specific training. During these early stages (training age 1–4 years), emphasis should be placed on developing athletic motor skills, such as jumping, landing, rebounding, and trunk muscle strength [3]. ST is particularly effective for enhancing these motor competencies, which subsequently support sport-specific performance in later LTAD stages [1].

Previous studies have primarily investigated the effects of lower- and upper-limb ST on physical fitness and rowing performance in young male and female rowers aged approximately 13 years [4,5]. For example, 9 weeks of heavy-resistance ST versus strength-endurance training showed greater gains in muscle strength (e.g., leg press, bench press 1-RM) and proxies of power (e.g., countermovement jump) following heavy-resistance ST, whereas strength-endurance training resulted in larger improvements in 700-m rowing ergometer performance. A systematic review and meta-analysis by the same author group [5] reported moderate effects of ST on lower-limb strength and rowing-specific performance across recreational, sub-elite, and elite rowers. However, in these studies, ST focused predominantly on the limbs rather than the trunk.

Even when ST primarily targets the limbs, trunk muscles are still activated to stabilize the torso and enable effective force transfer during multi-joint movements. While previous studies in young athletes have mainly targeted upper- and/or lower-limb muscles, trunk muscles play a central role in sports such as rowing, transferring forces from the lower limbs through a stable trunk to the upper limbs and ultimately to the oars to generate boat propulsion [5,6]. Anatomically and functionally, trunk muscles can be categorized as local or global [7]. While local muscles attach segmentally to the lumbar vertebrae and primarily provide intersegmental control and spinal stability, global muscles attach to the pelvis, ribcage, or long spinal lever arms, contributing to trunk torque generation, force transfer, and gross movement control [7]. From a biomechanical perspective, effective rowing performance relies on coordinated segmental contributions in which the trunk muscles facilitate efficient force transmission from the lower limbs to the arms, while sufficient shoulder mobility allows the generated forces to be effectively applied to the oar [8,9]. Hibbs et al. [8] proposed that global trunk muscle ST primarily enhances components of physical fitness such as speed, power, and agility, while local trunk muscle ST mainly supports trunk stability and injury prevention. These divergent responsibilities highlight the importance of targeting both local and global muscles within kinetic chains to optimize performance and reduce injury risk.

Previous research has largely focused on local trunk muscle ST in adults and recreational rowers, with limited evidence on global trunk muscle ST or its effects in youth athletes. Tse et al. [6] reported no significant improvements in jump, change-of-direction speed, or rowing performance after local trunk muscle ST in recreational rowers. In

contrast, Zinke et al. [7] demonstrated that eight weeks of isokinetic, global trunk muscle ST significantly increased trunk rotator peak torque and was strongly associated with paddle force during flume-based boat propulsion in world-class canoeists.

Despite these preliminary insights, the differential effects of global versus local trunk muscle ST on physical fitness and rowing-specific performance have not yet been examined in pubertal male rowers. Therefore, we aimed to contrast GST with LST to examine the effects of specific trunk muscle ST on lower- and upper-limb strength and power, trunk muscle strength, and rowing-specific performance in young rowers. Based on anatomical, functional, and contemporary training frameworks [7–11], we hypothesized that GST would produce greater improvements than LST in lower- and upper-limb strength and power, trunk muscle strength, and rowing-specific performance in pubertal male rowers. These effects were anticipated because GST targets the large trunk muscle groups responsible for force generation and intersegmental coordination, thereby enhancing kinetic chain efficiency and facilitating more effective power transfer from the lower limbs through a stiff, powerful trunk to the upper limbs. In contrast, LST primarily improves segmental stability without substantially increasing force output [8]. Based on previous literature [11,12] and using the Hopkins et al. [13] classification, we anticipated moderate effects (Cohen's $d = 0.5$–0.8) favoring GST for dynamic performance measures such as counter-movement jump and 700-m rowing ergometer performance.

## Methods

### Participants

Sample size was estimated using G*Power (version 3.1.6). Based on a related study [12] examining heavy-resistance versus strength-endurance training on bench pull strength in young rowers (Cohen's $f = 0.30$), an a priori power analysis with $\alpha = 0.05$ and 80% power indicated that 24 participants would be sufficient (Table 1). Accordingly, 28 pubertal male rowers were recruited from a youth rowing center. Participants were randomly assigned to the GST (n = 15) and LST (n = 13) groups using a computer-generated randomization sequence. Both groups continued their regular training program, which included an average of 20 hours per week of on-water rowing and 4 hours per week of off-water ergometer training, in line

**Table 1. Anthropometrics of the examined study cohort according to group allocation.**

|  | GST (n = 15) | LST (n = 13) | Independent sample t-test p-value |
|---|---|---|---|
| **Age (years)** | 13.7 ± 1.1 | 13.6 ± 0.2 | 0.705 |
| **Body height (cm)** | 163.8 ± 6.9 | 164.5 ± 12.9 | 0.812 |
| **Sitting height (cm)** | 78.0 ± 4.9 | 78.7 ± 6.2 | 0.942 |
| **Body mass (kg)** | 53.3 ± 7.9 | 53.1 ± 10.2 | 0.841 |
| **BF%** | 12.5 ± 1.2 | 12.3 ± 1.8 | 0.765 |
| **Maturity offset (years)** | 0.3 ± 0.2 | 0.2 ± 0.5 | 0.943 |
| **Predicted APHV (years)** | 13.6 ± 0.3 | 13.6 ± 0.5 | 0.836 |
| **Back extensor strength (kg)** | 71.5 ± 12.8 | 73.6 ± 12.0 | 0.970 |
| **1-RM bench press (kg)** | 33.7 ± 6.2 | 35.9 ± 12.0 | 0.789 |
| **1-RM half squat (kg)** | 59.3 ± 17.3 | 59.9 ± 10.8 | 0.867 |
| **CMJ-height (cm)** | 26.1 ± 3.9 | 27.6 ± 5.8 | 0.786 |
| **SLJ (cm)** | 199.8 ± 33.4 | 202.7 ± 25.9 | 0.798 |
| **Seated medicine ball throw (m)** | 3.6 ± 0.7 | 3.6 ± 0.5 | 0.876 |
| **Backward medicine ball throw (m)** | 5.5 ± 0.5 | 5.4 ± 1.3 | 0.954 |
| **700-m Rowing ergometer test (s)** | 199.6 ± 41.2 | 188.9 ± 48.6 | 0.851 |

**Notes:** Data are presented as means and standard deviations; BF: body fat; GST: global trunk muscle strength training; LST: local trunk muscle strength training; APHV: age at peak-height-velocity; m: meters; s: seconds; cm: centimetre; kg: kilogram.

with the planned training schedule. Participants were aged 12–13 years (circa-PHV = 0.2–0.3; age at PHV = 13.6 years) and classified as Tier 2 (trained/developmental) according to McKay et al. [14]. They had 4–5 years of general strength and conditioning experience, while their sport-specific training focused on rowing technique. Inclusion criteria were: male rowers aged 12–13 years, circa-peak height velocity, a minimum of 4 years of rowing training including general strength and conditioning, regular participation in at least four weekly training sessions, and no prior experience with systematic global or local trunk muscle ST interventions. Exclusion criteria included a history of musculoskeletal, neurological, or orthopedic disorders within six months prior to the study, cardiovascular or metabolic conditions limiting safe participation, absence from more than 10% of training or testing sessions, engagement in additional structured resistance training programs outside the study, or use of medications likely to affect neuromuscular performance. Prior to participation, athletes and their legal guardians received detailed information on study objectives, procedures, risks, and benefits, and provided written consent. The study adhered to the Declaration of Helsinki and was approved by the Local Clinical Research Ethics Committee (Personal Protection Committee; Code: N° 0227//2025). None of the participants reported psychological, musculoskeletal, neurological, or orthopedic disorders within six months before the study.

## Procedures

One week before the study, participants completed a familiarization session to become acquainted with all tests and exercises. Proper techniques for the GST and LST exercises were explained and practiced. Consistent with prior training studies in young athletes [15,16], a passive control group was not included, as with holding training during the pre-season would have conflicted with ethical and practical considerations. However, alternative designs such as attention- or activity-matched control groups could be implemented to control for training exposure and participant contact. Given that previous research has already established the general effectiveness of trunk muscle ST in youth athletes [17], the primary aim of the present study was to compare the specific effects of GST versus LST on physical fitness and rowing-specific performance. Pre- and post-training assessments included lower-limb strength and power (1-RM half squat, countermovement jump [CMJ], standing long jump [SLJ]), upper-limb strength and power (1-RM bench press, seated and backward medicine ball throw), and a 700-m rowing ergometer test. Before testing, participants completed a standardized warm-up consisting of 5 minutes of submaximal running, 2–3 submaximal sprints over 10–15 m, and rowing-specific exercises (Fig 1).

The sprint distance was selected to mimic the short, explosive movements characteristic of the start and drive phases in rowing, while keeping the intensity submaximal to avoid fatigue prior to testing. The best trial of each test was used for analysis, with the same sequence applied at pre- and post-tests.

## Anthropometrics and maturity

Athletes' body height and mass were measured using a wall-mounted stadiometer (Florham Park, NJ) and an electronic scale (Baty International, West Sussex, England), respectively. Body composition was assessed via the sum of 4 skinfolds measured at the biceps, triceps, subscapular and suprailiac using Harpenden skinfold calipers. All measurements were performed by a certified ISAK Level 2 anthropometrist to ensure reliability. Anthropometric testing followed the protocol described by Deurenberg et al. [18], which reported similar prediction errors for adults and adolescents. Biological maturity was then estimated non-invasively using chronological age, standing height, and sitting height in a validated regression equation to predict maturity offset [19], with a standard error of estimate of 0.542 years for boys.

### Muscle strength and power assessment

**Back extensor strength.** Maximal isometric back extensor strength was measured in kilograms using a back and leg dynamometer (Takei, Tokyo, Japan) as previously described [20]. Athletes stood on the force plate with feet shoulder width apart and gripped the handle bar positioned across the thigh. The chain length was adjusted so that the legs remained straight while participants flexed at the hip by approximately 30°, resulting in a forward trunk inclination that positioned

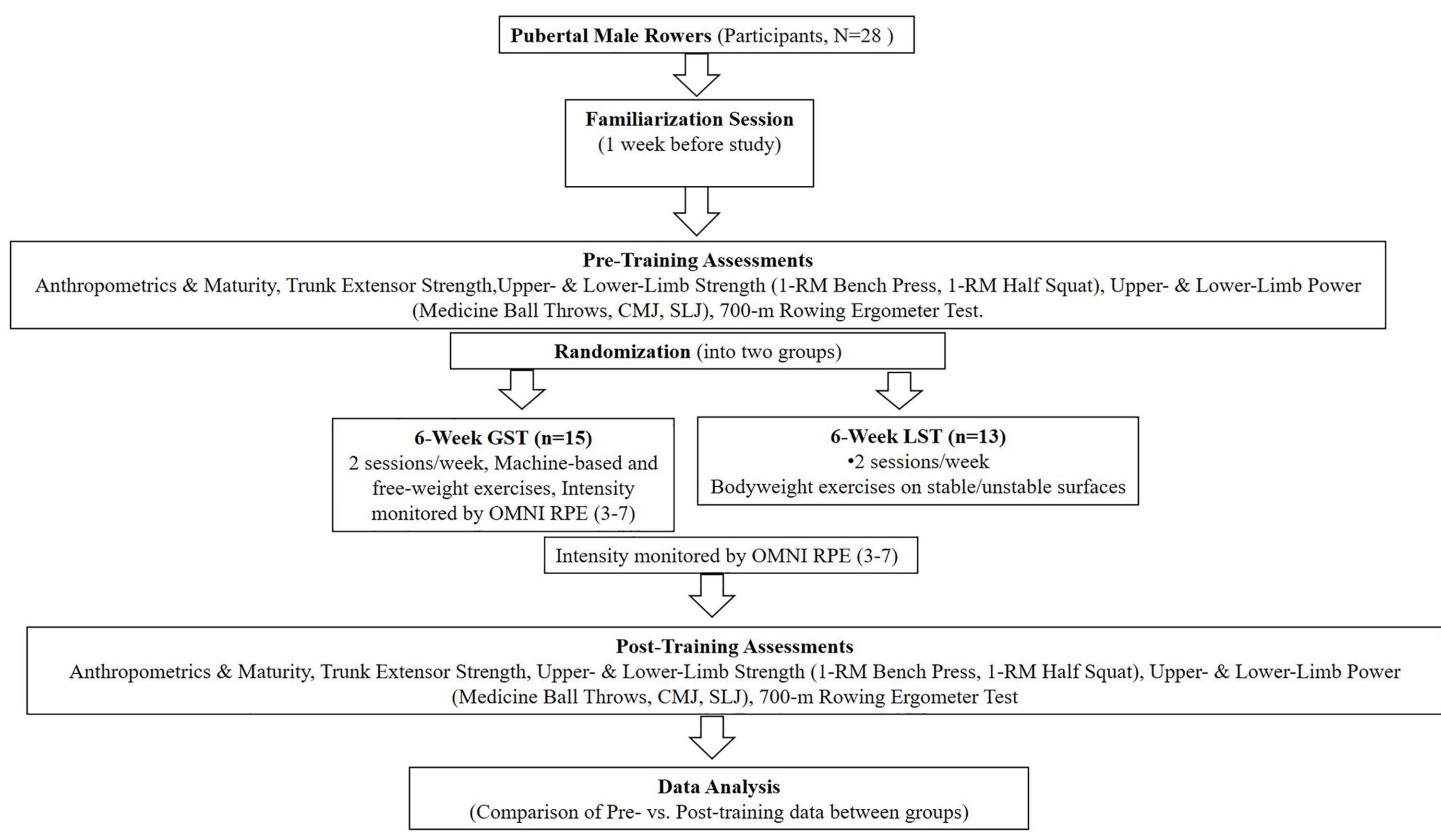

**Fig 1. Experimental design. Twenty-eight pubertal male rowers were randomized to 6 weeks of (GST; n = 15) or (LST; n = 13), performed twice weekly.** Pre- and post-training assessments included anthropometrics and maturity status, trunk extensor strength, upper- and lower-limb maximal strength (one-repetition maximum [1RM] bench press and half squat), upper- and lower-limb power (medicine ball throws, countermovement jump [CMJ], standing long jump [SLJ]), and a 700 m rowing ergometer test. Training intensity was monitored using the OMNI rating of perceived exertion scale (OMNI-RPE; 3–7). Pre- to post-training changes were compared between groups.

the bar at the level of the patella. Participants were then asked to straighten (extend) the back (i.e., stand upright) without bending their knees and to lift the chain, with the pulling force applied to the handle. The athletes were asked to pull as forcefully as possible. The participants completed three trials and the best trial was used for further analysis. A thirty-second rest interval was provided between trials. While this test demonstrates excellent test-retest reliability (ICC = 0.98, SEM = 1.18%) in adult populations [20], its responsiveness to short-term training interventions in pubertal athletes may be influenced by maturational factors, motivational variability, and day-to-day performance fluctuations [21]. Observed changes should therefore be interpreted within this context. However, when applied to youth populations and short-term interventions, broader methodological considerations are necessary. In particular, test responsiveness, i.e., the ability to detect meaningful changes resulting from training may be influenced by maturation, motivation, and day-to-day variability [22]. Therefore, while the test is highly reliable, observed changes should be interpreted in the context of these factors when assessing training-induced adaptations over a six-week period.

**Upper and lower limb muscle strength.** Upper (1-RM bench press) and lower limb (1-RM half squat) strength tests were performed following standardized procedures as described by Keiner et al. [23] and in line with recent methodological recommendations for male athletes [21]. A standardized protocol was applied, beginning with three submaximal sets of 1–5 repetitions at light-to-moderate loads adjusted for body mass [24]. Participants then performed

three series of two repetitions each at progressively heavier loads corresponding to approximately 70% and 80% of estimated 1-RM. Load increments of 1–2 kg were applied based on the athlete's capacity, with progressively smaller increases as participants approached their 1-RM. Failure was defined as inability to complete the full range of motion on at least two attempts, with two minutes of rest between trials. The 1-RM was typically determined within 6–8 attempts. All testing was conducted with an instructor-to-athlete ratio of 1:1. This protocol accounts for body mass considerations and allows for accurate 1-RM prediction in male athletes [21]. Excellent test–retest reliability has been reported for upper and lower limb 1-RM tests (ICCs = 0.87; SEM = 4.52%) [25].

### Proxies of muscle power

**Medicine ball throw test.** The medicine ball throw test was used to assess upper-body power previously showing high reliability in young athletes with an ICC = 0.96 and SEM = 5.4% [26]. During testing, participants sat on the ground with their legs extended while keeping both hands on their chest and their back against a wall. Athletes were then instructed to push a medicine ball of 3 kg as forcefully as possible in a horizontal direction. The examiner visually verified the correct pushing technique. The distance from the wall to the point where the medicine ball landed was recorded and the best out of three test trials was used for further statistical analysis.

**Back medicine ball throw test.** The test was performed in an erect standing position according to the description provided by Stockbrugger and Haennel [27] using a 27-cm diameter, 7-kg rubber medicine ball, with a granulated surface for easy gripping. A brief description of the proper test execution was provided to each athlete based on the suggested angle of release to achieve maximum horizontal distance [28]. During testing, the athletes stood at the baseline of an indoor basketball court and grasped the medicine ball with both hands. Knees were slightly flexed and the ball was lowered to approximately knee height. From there, athletes extended their legs as forcefully and rapidly as possible, followed by a back extension, elevation of the shoulders, and finally shoulder flexion to throw the ball backward over the head in an effort to achieve maximum horizontal distance. Each athlete performed three trials, separated by approximately three minutes of recovery [29]. The best throw was used for further analysis. Previously, test-retest reliability was described as excellent with an ICC = 0.96 and SEM = 0.33% [30].

**Countermovement jump test (CMJ).** The CMJ was assessed using the Ergojump system following the procedures described by Hammami et al. [15]. The device calculates flight time and subsequently derives jump height using standard kinematic equations. As the system is technically not capable of recording kinetic data, i.e., force–time, peak forces, impulses, or rate of force development we were unable to report mechanical outcomes. Therefore, jump height was used as the primary CMJ outcome parameter.

Athletes stood upright, flexed their legs to a self-selected knee angle, and extended explosively in the vertical direction. Three trials were performed with approximately two minutes of recovery between trials, and the best result was used for analysis. Excellent test-retest reliability has been reported for jump height, with an ICC of 0.92 and SEM of 2.54% [15]. Of note, jump height is a simple, reliable, and highly reproducible measure in young athletes and is commonly used in field-based testing, allowing for meaningful comparisons with previous research [20].

**Standing long jump test (SLJ).** The SLJ was performed according to previously described procedures by Hammami et al. [15]. During testing, athletes in upright erect position and feet shoulder width apart behind the starting line. They then extended their legs as forcefully and rapidly as possible to jump in horizontal direction. The distance was measured from the starting line to the point where the back of the heel landed. The test was repeated twice, and the best score in cm was used for further analysis [31]. Previously, test retest reliability was excellent with an ICC = 0.92 and SEM = 1.66% [31].

### Rowing specific ergometer test

A 700-m all-out rowing ergometer test was performed using a Concept II ergometer (Model D, Morrisville, Vermont, United States) to assess rowing-specific performance [12]. The test was conducted in accordance with previously published

guidelines by Thiele et al. [12]. A drag factor of 120 was applied, corresponding to a fan resistance setting of 3, which is appropriate for junior athletes at this performance level. Test-retest reliability for the 700-m ergometer trial has been reported as excellent (ICC = 0.99, SEM = 0.7%) [12,32]. Stroke rate was self-selected, and athletes were instructed to exert maximal effort throughout the entire test. Average power output was recorded for further analysis. The warm-up protocol included 15 minutes of running or cycling followed by 15 minutes of submaximal rowing on the ergometer.

**Global versus local trunk muscle strength training.** The applied ST programs were designed following the LTAD periodization strategies proposed by Haff [2]. Both groups completed a 6-week pre-season trunk muscle ST program, with two sessions per week in addition to regular rowing training focused on technical skill development [26]. Each 90-minute session included a standardized 15-minute warm-up, 35 minutes of rowing exercises emphasizing technique, 30 minutes of trunk muscle ST (GST or LST), and a 10-minute cool-down.

GST participants performed machine-based and free-weight exercises targeting ventral, lateral, and dorsal trunk muscles at ~70% of the 1-RM, emphasizing progressive overload to increase force production. LST participants performed exercises with the own body mass (e.g., planks, bird dogs) using stable and unstable surfaces (Swiss ball, BOSU) to target the deep-lying trunk muscles [33], focusing on intersegmental stability and motor control. A notable methodological consideration is that the comparison between GST and LST is confounded by differences in external loading (70% 1-RM vs. body mass exercises), which precludes definitive attribution of observed effects to muscle group targeting per set. We acknowledge that the comparison between high-load GST and LST using the own body mass represents an inherent limitation, as differences in external load may confound the effects of exercise type. To mitigate this, both interventions were structured with systematic progression in perceived exertion (RPE) and exercise complexity to provide comparable relative training challenges within each modality. Exercise selection was guided by the functional roles of the trunk muscles. ST of global muscles targeted force generation and transfer (GST). ST of local muscles segmental stability and control (LST) (Table 2). During the 6-week training period, session ratings of perceived exertion (sRPE) were collected after each training session using the validated 0–10 OMNI scale and we targeted to progressively increase sRPE over the course of the training study. At program initiation (weeks 1–2), participants were instructed to maintain an sRPE of approximately 3. For weeks 3–4, the target sRPE was increased to 5, and for weeks 5–6, to 7, following a structured, linear progression

**Table 2. Design of the global (GST) versus local (LST) trunk muscle strength training.**

| Exercises | Week 1 | Week 2 | Week 3 | Week 4 | Week 5 | Week 6 |
|---|---|---|---|---|---|---|
| Local trunk muscle ST comprised planks using the own body mass on stable and unstable surfaces (Swiss balls, BOSU etc). | | | | | | |
| Frontal dynamic trunk frontal on Swiss ball | 3 × 20 (s) | 3 × 30 (s) | 3 × 40 (s) | 3 × 20 (s) | 3 × 40 (s) | 3 × 50 (s) |
| Dorsal dynamic trunk on Bosu (left and right) | 3 × 20 (s) | 3 × 30 (s) | 3 × 40 (s) | 3 × 20 (s) | 3 × 40 (s) | 3 × 50 (s) |
| Lateral dynamic trunk on Swiss ball | 3 × 20 (s) | 3 × 30 (s) | 3 × 40 (s) | 3 × 20 (s) | 3 × 40 (s) | 3 × 50 (s) |
| Global trunk muscle ST using machine-based or free weight exercises at an intensity of 70% of the 1-RM for ventral, lateral, dorsal trunk muscles. | | | | | | |
| Cable modified deadlift | 3 × 10 reps at (70% 1-RM) | 3 × 15 reps (70% 1-RM) | 3 × 20 reps (70% 1-RM) | 3 × 10 reps (70% 1-RM) | 3 × 20 reps (70% 1-RM) | 3 × 25 reps (70% 1-RM) |
| Latera smith machine shrug | 3 × 10 reps at (70% 1-RM) | 3 × 15 reps (70% 1-RM) | 3 × 20 reps (70% 1-RM) | 3 × 10 reps (70% 1-RM) | 3 × 20 reps (70% 1-RM) | 3 × 25 reps (70% 1-RM) |
| Cable incline pulldown | 3 × 10 reps at (70% 1-RM) | 3 × 15 reps (70% 1-RM) | 3 × 20 reps (70% 1-RM) | 3 × 10 reps (70% 1-RM) | 3 × 20 reps (70% 1-RM) | 3 × 25 reps (70% 1-RM) |

**Notes:** ST: strength training; GST: global trunk muscle strength training; LST: local trunk muscle strength training; 1-RM: one-repetition-maximum.; reps: repetition.

approach. Mean sRPE across the intervention was 6.5±0.4 for GST and 6.0±0.3 for LST. Participants' adherence rate to training was recorded throughout the study period.

## Statistical analyses

Data are presented as means and standard deviations (SD). Data normality were assessed and confirmed using the Shapiro-Wilk test. To examine the effects of the two training programs on dependent variables, a 2 (groups: GST, LST) × 2 (times: pre-, post-test) repeated-measures ANOVA was computed. To account for inter-individual differences in biological maturation, maturity offset was included as a covariate in the ANOVA analyses. This approach allowed to elucidate the effects of training and time to be examined independently of maturity-related variability. Where applicable, adjusted means and interaction effects were therefore reported after accounting for PHV. When a significant group-by-time interaction was observed, post-hoc paired t-tests were performed with Bonferroni correction to account for multiple comparisons across outcome measures.

Effect sizes (ES) were calculated separately for within-group and between-group comparisons. Within-group changes were quantified using Cohen's $d$, calculated as the mean difference divided by the pooled SD of pre- and post-values ($d$ = [M_post – M_pre]/ SD_pooled). Between-group effect sizes for change scores were calculated as Cohen's $d$ using the difference in change scores divided by the pooled SD. Partial eta-squared values derived from ANOVA were converted to Cohen's $d$ to ensure uniform reporting and transparency [34]. Effect sizes were interpreted according to Hopkins et al. [13] as trivial (<0.2), small (0.2–0.6), moderate (0.6–1.2), large (1.2–2.0), very large (2.0–4.0), and extremely large (>4.0). The significance level was set at $p < 0.05$. All analyses were performed using SPSS 25.0 (SPSS Inc., Chicago, IL, USA).

# Results

All participants received treatments as allocated. No training or test-related injuries occurred over the course of the study. Group specific data on anthropometrics and body composition at baseline are displayed in Table 1. No statistically significant between-group baseline differences were observed with respect to anthropometrics, body composition, biological age (maturity offset), physical fitness, and rowing-specific performance (Table 1).

## Measures of muscle strength

**Back extensor strength.** The analysis indicated a significant group-by-time interaction for back extensor strength ($d$=3.48 [very large], $p < 0.001$) (Table 3). Post-hoc analyses showed large pre- to post-test performance improvements for the GST (Δ22.5%; $p < 0.0001$; $d$=1.22 [CI for mean difference: −8.17 to 6.78]) and a smaller improvement for the LST (Δ8.3%; $p < 0.001$; $d$=0.52 [CI for mean difference: −6.59 to 5.65]).

**Upper and lower limb muscle strength.** Results indicated a significant group-by-time interaction ($d$=3.00 [very large], $p < 0.001$) for 1-RM bench press (Fig 2). Post-hoc analyses demonstrated a large performance enhancement from pre- to post-test for GST (Δ25.1%; $p < 0.001$; $d$=1.37 [CI for mean difference: −4.75 to 2.21] and a smaller improvement for LST (Δ7.2%; $p < 0.001$; $d$=0.22 [CI for mean difference: −6.29 to 6.36]).

A significant group-by-time interaction ($d$=3.91 [very large], $p < 0.001$) was found for the 1-RM half squat (Table 3). Post-hoc analyses demonstrated a moderate performance improvement from pre- to post-test for the GST (Δ17.3%; $p < 0.001$; $d$=0.61 [CI for mean difference: −10.01 to 8.90]) and a small improvement for LST (Δ7.3%; $p < 0.001$; $d$=0.43 [CI for mean difference: −5.90 to 4.83]).

## Proxies of upper limb muscle power

**Medicine ball throw test.** For the seated medicine ball throw test, a significant group-by-time interaction ($d$=0.81 [medium], $p < 0.001$) was observed. Post-hoc analyses indicated a small improvement for the GST (Δ8.0%; $p < 0.05$;

**Table 3. Group-specific means and standard deviations for all outcome measures before (pre) and after (post) the intervention period.**

| | LST (n = 15) | | | | Δ % | GST (n = 13) | | | | Δ % | ANOVA | |
| | Pretest | | Posttest | | | Pretest | | Posttest | | | p-value (d) | |
| | M | SD | M | SD | | M | SD | M | SD | | Time | Group × Time |
|---|---|---|---|---|---|---|---|---|---|---|---|---|
| **Proxies of muscle strength** | | | | | | | | | | | | |
| *Back extensor strength (kg)* | 73.6 | 12 | 79.7 | 12.2 | 8.3 | 71.5 | 12.8 | 87.6 | 14.7 | 22.5 | 0.001 (6.88) | 0.001 (3.48) |
| *1-RM bench press (kg)* | 35.8 | 12 | 38.4 | 13 | 7.2 | 33.7 | 6.2 | 42.2 | 6.6 | 25.1 | 0.001 (5.00) | 0.001 (3.00) |
| *1-RM half squat (kg)* | 59.9 | 10.8 | 64.3 | 10.4 | 7.3 | 59.3 | 17.3 | 69.5 | 17.5 | 17.3 | 0.001 (8.81) | 0.001 (3.91) |
| **Proxies of muscle power** | | | | | | | | | | | | |
| *CMJ-height (cm)* | 27.6 | 5.8 | 29.1 | 5.8 | 5.5 | 26.1 | 3.9 | 32.5 | 3.1 | 24.3 | 0.001 (3.64) | 0.001 (2.43) |
| *SLJ (cm)* | 202.7 | 25.9 | 209.9 | 27.3 | 3.5 | 199.8 | 33.4 | 215.7 | 34.2 | 7.9 | 0.001 (2.81) | 0.01 (1.19) |
| *Seated medicine ball throw (m)* | 3.6 | 0.5 | 3.8 | 0.5 | 3.2 | 3.6 | 0.7 | 21 | 6.1 | 8 | 0.001 (1.76) | 0.06 (0.81) |
| *Back medicine ball throw (m)* | 5.4 | 1.3 | 5.7 | 1.2 | 4 | 5.5 | 0.5 | 5.9 | 0.5 | 8.2 | 0.001 (3.73) | 0.01 (1.45) |
| **Rowing-specific performance** | | | | | | | | | | | | |
| *700-m rowing test (s)* | 188.9 | 48.6 | 183.8 | 47.5 | 2.7 | 199.6 | 41.2 | 189 | 38.4 | 5.3 | 0.001 (4.05) | 0.001 (1.61) |

**Notes:** M: mean; SD: standard deviation; GST: global trunk muscle strength training; LST: local trunk muscle strength training: one maximum repetition; CMJ: countermovement jump; SLJ: standing long jump; d = Cohen's d. Confidence intervals represent 95% CIs for mean differences in raw score units.

d = −0.44 [CI for mean difference: −0.80 to −0.09]) and a smaller improvement for the LST (Δ3.2%; p < 0.001; d = −0.41 [CI for mean difference: −0.67 to −0.16]). For the back medicine ball throw test, a significant group-by-time interaction (d = 1.45 [large], p < 0.01) was identified. Post-hoc analyses indicated large improvements for the GST group (Δ8.2%; p < 0.001; d = −0.83 [CI for mean difference: −1.10 to −0.56]) and slightly smaller improvements for the LST (Δ4.0%; p < 0.001; d = −0.25 [CI for mean difference: −0.91 to 0.36]).

**Countermovement jump test.** For CMJ-height, a significant group-by-time interaction (d = 2.43 [large], p < 0.001) was observed (Fig 3). Post-hoc analyses demonstrated large CMJ improvements for the GST (Δ24.3%; p < 0.001; d = −1.89 [CI for mean difference: −4.01 to −0.21]) and small improvements for the LST (Δ5.5%; p < 0.001; d = −0.27 [CI for mean difference: −3.20 to 2.67]).

**Standing long jump test.** For the SLJ test, a significant group-by-time interaction (d = 1.19 [large], p < 0.05) was noted. Post-hoc tests indicated a large pre-post improvement for the GST (Δ7.9%; p < 0.001; d = −0.49 [CI for mean difference: −18.56 to 18.10]) and a small improvement for the LST (Δ3.5%; p < 0.001; d = −0.28 [CI for mean difference: −13.39 to 13.54]).

## Rowing-specific performance

For the 700-m rowing test, a significant group-by-time interaction (d = 1.61 [large], p < 0.001) was found (Fig 4). Post-hoc tests showed a small improvement for GST (Δ5.3%; p < 0.001; d = −0.28 [CI for mean difference: −22.12 to 21.15]) and a trivial improvement for LST (Δ2.7%; p < 0.001; d = 0.11 [CI for mean difference: −24.48 to 24.20]).

## Discussion

This study investigated the effects of a 6-week pre-season global (GST) versus local (LST) trunk muscle ST on measures of physical fitness and sport-specific performance in pubertal male rowers with Tier 2 training and performance caliber. The main findings showed that both programs produced significant gains in strength, proxies of power, and rowing-specific performance, with greater improvements observed in the GST group. These effects likely reflect the combined influence of exercise modality (GST versus LST) and higher external loading intensity during GST, rather than training modality alone. Therefore, interpretations of GST's apparent superiority should be made with caution.

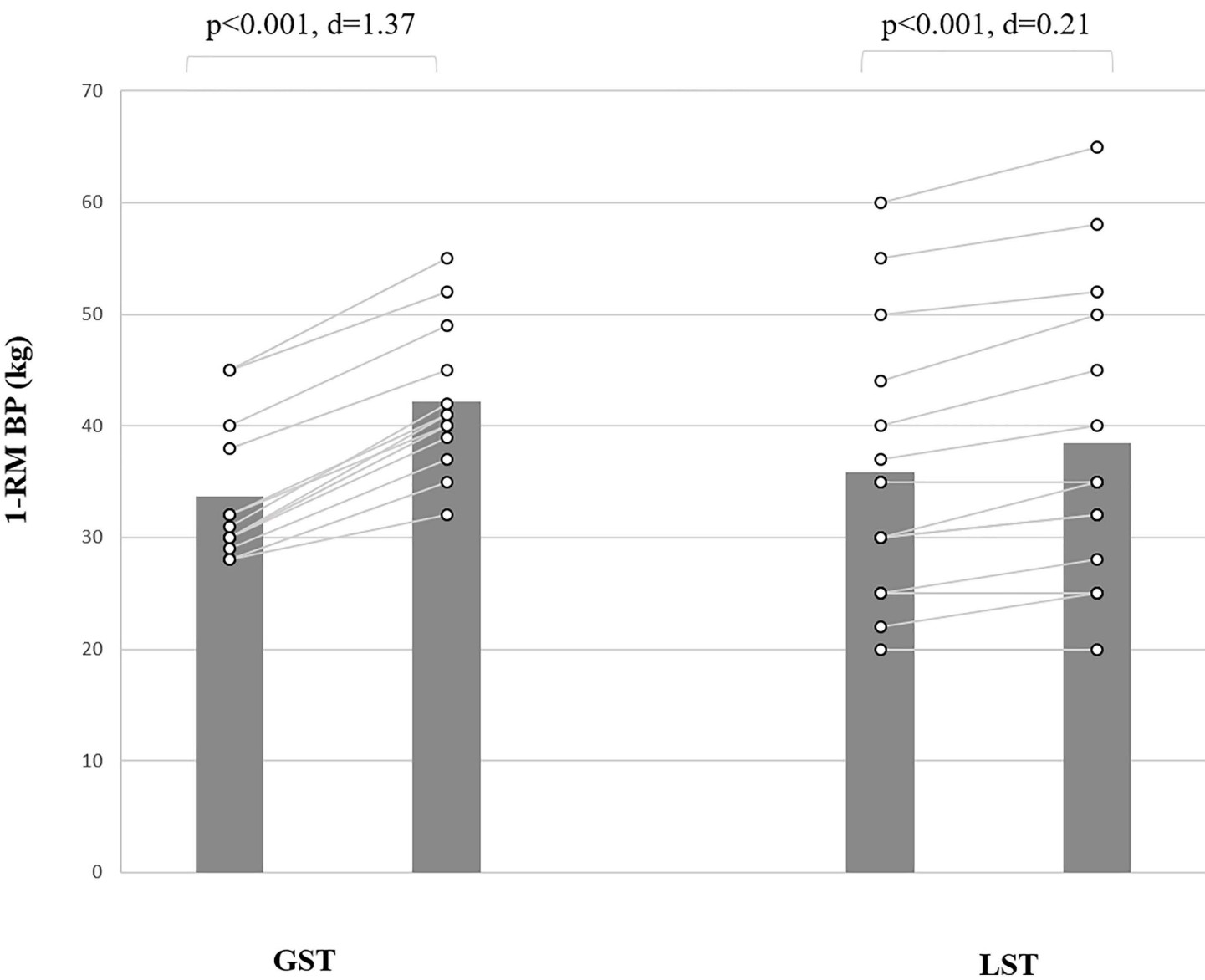

**Fig 2. The figure shows training-induced adaptations that occurred in 1-RM bench press performance in male pubertal rowers.** GST: global trunk muscle strength training; LST: Local trunk muscle strength training; M: mean; SD: standard deviation; 1-RMBP: one-repetition-maximum bench press.

Only a few studies have examined the impact of trunk muscle ST on performance development in young rowers [7,12]. For instance, Thiele et al. [12] reported greater improvements in muscle strength and proxies of power following heavy-resistance ST compared to strength-endurance training in female Tier 3 rowers. Similarly, Held et al. [35] found that both

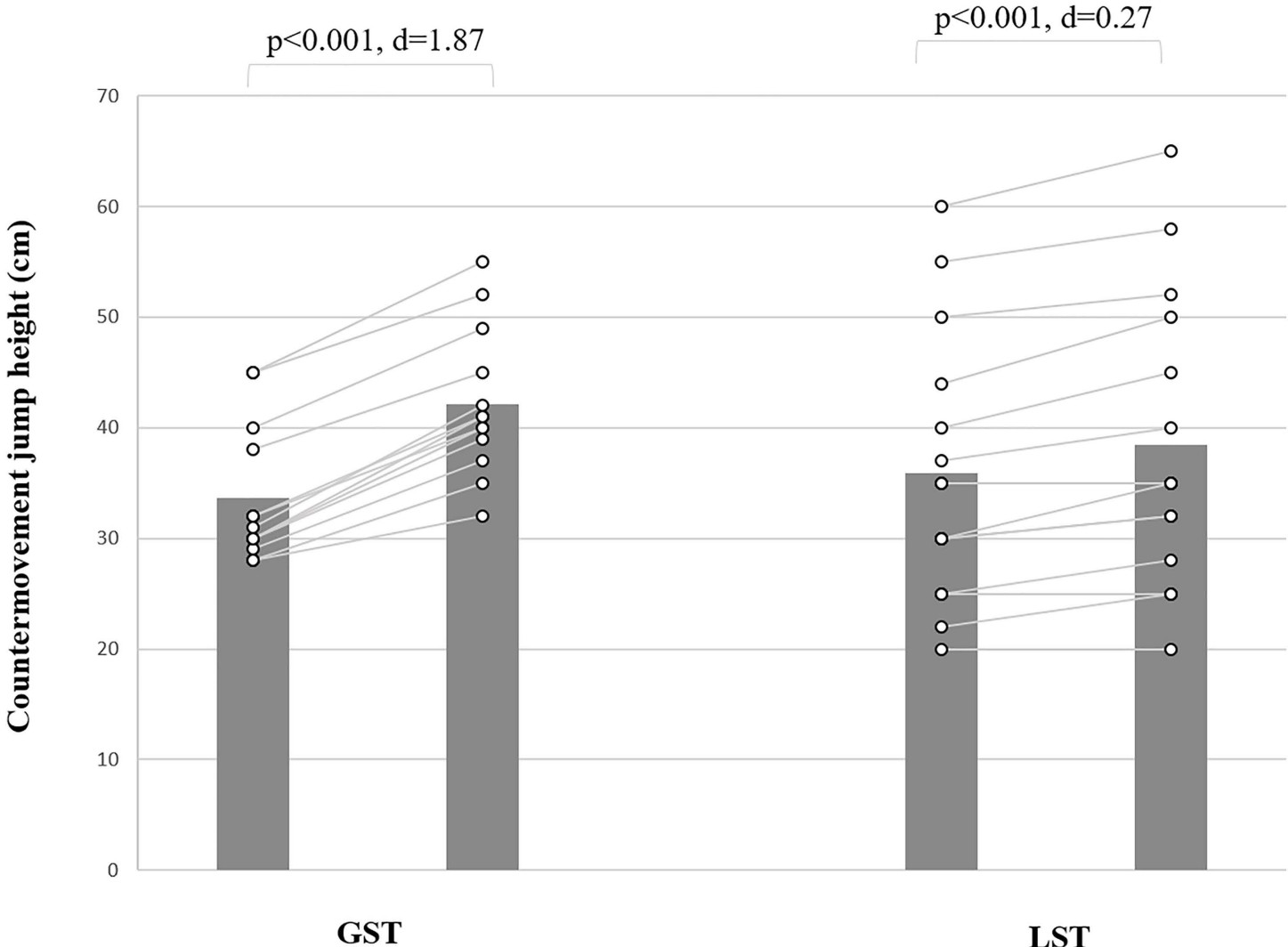

**Fig 3. The figure shows training-induced adaptations that occurred in countermovement jump height in male pubertal rowers.** GST: global trunk muscle strength training; LST: local trunk muscle strength training; M: mean; SD: standard deviation.

heavy-resistance ST and strength-endurance training improved 2,000-m ergometer performance in female rowers, while Naghizadeh et al. [36] observed superior performance outcomes after high-intensity interval training compared to traditional ST. In related water sports, Zinke et al. [7] demonstrated that isokinetic (global) trunk muscle ST enhances trunk rotator torque, which is strongly associated with paddle force during flume-based boat propulsion in world-class canoeists. Collectively, these studies suggest that higher ST intensities which primarily target global trunk muscles may yield greater performance gains, regardless of the specific training modality used.

In rowing, the specific contribution of trunk muscle ST remains insufficiently understood. While Tse et al. [6] found no significant improvements in rowing performance after LST, Simon et al. [37] observed enhanced trunk stability and

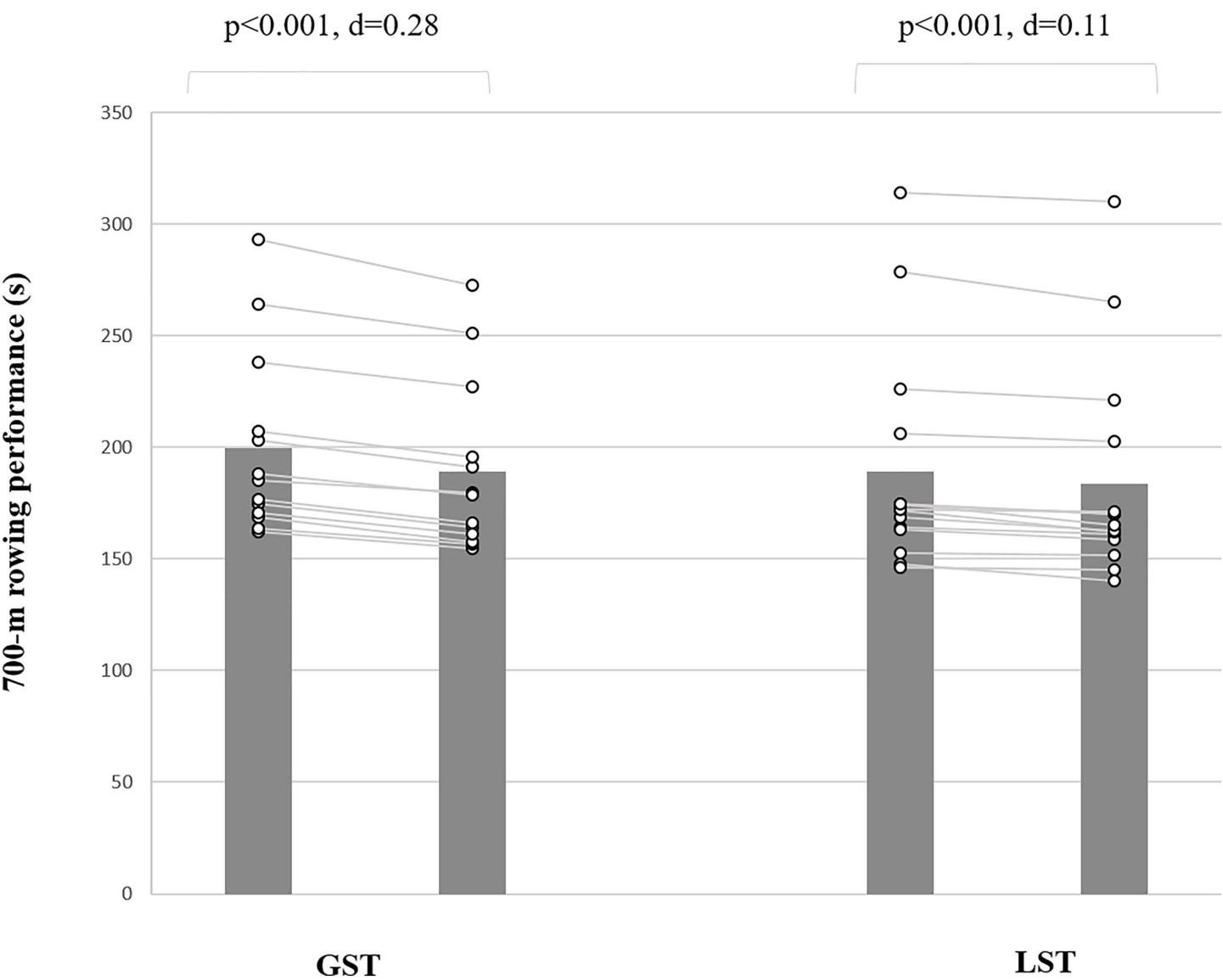

**Fig 4. The figure illustrates training-induced adaptations that occurred in 700-m rowing performance in male pubertal rowers.** GST: global trunk muscle strength training; LST: local trunk muscle strength training; M: mean; SD: standard deviation.

improved ergometer performance following GST. In their meta-analysis, Saeterbakken et al. [17] reported small-to-large effects of trunk muscle ST on muscular fitness and sport-specific outcomes, with the greatest benefits observed in younger athletes. However, these authors did not distinguish between ST for the global or local trunk muscles, further underscoring the need for studies that control for both, training intensity and modality.

The present results align with the conceptual trunk muscle model proposed by Hibbs et al. [8], in which GST supports whole-body force production and performance enhancement while LST contributes to postural stability and injury prevention. In the current study, GST involved externally loaded exercises (~70% 1-RM), whereas LST relied on exercises using the own body mass. Therefore, the observed performance differences cannot be attributed solely to global versus local trunk muscle recruitment, as the higher training load in GST likely contributed to adaptations in strength, power, and rowing-specific performance. Although instability in LST may have increased trunk muscle activation [9,33,38], this remains speculative due to the applied methods and does not resolve the confounding between exercise modality and load. Consequently, mechanistic interpretations regarding force transmission or global muscle recruitment should be made cautiously.

In terms of sport-specific outcomes and in accordance with Hibb et al. [8] conceptual model, GST produced larger improvements than LST in a 700-m rowing ergometer trial among Tier 2 pubertal rowers likely reflecting the combined influences of enhanced trunk muscle strength and the higher external loading employed in GST. These gains most likely stem from the greater external resistance or the specific activation of the global trunk muscles used in GST, which would have provided a more effective stimulus for neuromuscular adaptation. While LST may remain valuable for addressing muscular imbalances or as part of rehabilitation and early developmental programs [12], future research should employ load-equated designs to isolate the effects of training modality (GST versus LST). Moreover, future studies combining GST and LST within periodized programs, incorporating both sexes and larger samples, would further enhance the generalizability of these findings. Finally, the application of biomechanical testing apparatus such as surface electromyography, motion analysis, and muscle imaging could clarify how different trunk muscle ST strategies contribute to rowing-specific performance.

Finally, training adaptations observed in Tier 2 pubertal rowers should be interpreted within the LTAD framework which highlights heightened neuromuscular responsiveness around PHV [39]. In the present study, improvements in muscle strength and power following GST likely reflect enhanced force transmission and increased maximal force production. Accordingly, ST in general, and GST in particular may be prescribed to accommodate individual differences in growth and maturation, thereby supporting the development of movement competencies during mid-PHV [1]. Nevertheless, these findings are specific to pubertal male rowers, and caution is warranted when extrapolating the current results to other populations (e.g., female athletes) or sport disciplines (e.g., canoeing).

## Study limitations

Several methodological limitations of this study must be acknowledged. First, no passive or active controls were included. Although ethical and logistical constraints limited this possibility, the absence of a control group restricts the ability to distinguish training effects from natural development or sport-specific practice. Second, and most critically, the confounding between exercise modality and training intensity represents a major design limitation. GST employed externally loaded exercises at approximately 70% of 1-RM, while LST relied on exercises using the own body mass. Although sRPE was systematically monitored to quantify internal training load and to verify potential between-group differences [40,41], no significant differences in ratings of sRPE and no indication of baseline between group differences were observed across the intervention period. Consequently, a key prerequisite for including sRPE as a covariate was not met, and it was therefore not incorporated into the inferential statistical model nor examined as a mediator or moderator of training effects. Future studies with larger samples and multiple assessment points could integrate internal load within longitudinal or mixed-effects frameworks to better elucidate dose-response relations [42].

Third, a key methodological limitation of this study is the use of the Ergojump system, which provides only flight-time derived jump height and does not capture force-time characteristics. Consequently, we were unable to analyze additional mechanical variables such as peak force, rate of force development, impulse that could have offered a more comprehensive evaluation of CMJ performance. Future studies should employ equipment (i.e., force plates) capable of recording full force-time profiles to enable more detailed assessments of neuromuscular performance.

Fourth, the a priori power analysis used bench pull strength as the reference outcome, as it represents upper-body pulling capacity and is functionally related to trunk muscle engagement during rowing, serving as a proxy for anticipated changes in trunk strength and rowing performance. However, because bench pull strength was not directly measured as a primary outcome in this study, this choice may limit the accuracy of the sample size estimation and should be considered when interpreting the findings.

Finally, Given the balanced design, the presence of only two time points, the absence of missing data, and no indication of baseline between-group differences, a 2 × 2 repeated-measures ANOVA was retained, which remains appropriate in this context despite biological heterogeneity in pediatric samples [43,44]. In addition, linear mixed-effect models (LMMs) were computed using the same dataset and yielded results comparable to those obtained with the ANOVA approach. Accordingly, we elected to retain the originally selected ANOVA framework. Nevertheless, future studies involving larger samples, additional time points, or greater developmental heterogeneity should preferentially adopt LMM to better capture individual growth trajectories, in line with contemporary recommendations in pediatric sport science [45].

## Conclusions

Six weeks of high-intensity GST produced greater improvements in muscle strength, proxies of power, and rowing-specific performance compared to LST in Tier 2 pubertal male rowers. However, these effects likely reflect the combined influences of exercise modality and loading intensity. Importantly, no training-related injuries were reported, indicating that both approaches were safe and well tolerated. Consequently, the current results cannot explicitly isolate the independent contribution of GST versus LST.

From a practical standpoint, coaches may consider incorporating GST with progressive external loading as part of foundational strength development programs for performance development and LST for promoting trunk stability and injury prevention. Nonetheless, future research employing load-equated or matched-intensity designs are needed to determine whether the observed effects are attributable to exercise modality, intensity, or their interaction.

## Supporting information

**S1 File. Rowing data-Plos One.**
(XLSX)

## Acknowledgments

The authors would like to acknowledge the contribution of Mr. Dhiaeddine Zoghlami for his extensive efforts in data collection for this study.

## Author contributions

**Conceptualization:** Raouf Hammami, Urs Granacher.

**Data curation:** Raouf Hammami, Abdelkader Mahmoudi.

**Formal analysis:** Yassine Negra, Walid Selmi.

**Funding acquisition:** Urs Granacher.

**Investigation:** Raouf Hammami, Abdelkader Mahmoudi.

**Methodology:** Raouf Hammami, Abdelkader Mahmoudi.

**Project administration:** Urs Granacher.

**Software:** Yassine Negra, Walid Selmi.

**Supervision:** Haithem Rebai, David Behm, Anis Chaouachi.

**Validation:** Anis Chaouachi, Urs Granacher.

**Visualization:** Walid Selmi, David Behm, Urs Granacher.

**Writing – original draft:** Raouf Hammami.

**Writing – review & editing:** Raouf Hammami, Haithem Rebai, Anis Chaouachi, Urs Granacher.

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
