## [Decision Letter · Decision Letter 0]

2 Sep 2025

Dear Dr. Hammami,

Thank you for submitting your manuscript to PLOS ONE. After careful consideration, we feel that it has merit but does not fully meet PLOS ONE’s publication criteria as it currently stands. Therefore, we invite you to submit a revised version of the manuscript that addresses the points raised during the review process.

We look forward to receiving your revised manuscript.

Kind regards,

Mário Espada, PhD

Academic Editor

PLOS ONE

Journal Requirements:

2. Please amend your authorship list in your manuscript file to include author David Behm.

3. Please amend the manuscript submission data (via Edit Submission) to include author David G Behm.

4. Please remove all personal information, ensure that the data shared are in accordance with participant consent, and re-upload a fully anonymized data set.

Additional Editor Comments :

Dear Authors,

Please revise the manuscript considering the reviewers´ suggestions.

Thank you.

Best regards.

Reviewers' comments:

Reviewer's Responses to Questions

**Comments to the Author**

1. Is the manuscript technically sound, and do the data support the conclusions?

Reviewer #1: Partly

Reviewer #2: Yes

Reviewer #3: Partly

2. Has the statistical analysis been performed appropriately and rigorously?

Reviewer #1: Yes

Reviewer #2: Yes

Reviewer #3: Yes

3. Have the authors made all data underlying the findings in their manuscript fully available?

Reviewer #1: Yes

Reviewer #2: Yes

Reviewer #3: Yes

4. Is the manuscript presented in an intelligible fashion and written in standard English?

Reviewer #1: Yes

Reviewer #2: No

Reviewer #3: Yes

Reviewer #1: Dear Authors,

I want to express my gratitude for the opportunity to review this manuscript.

The topic is relevant to this area of research, but the document requires improvement.

Below are suggestions with line indication:

Abstract:

36-60 – Please consider reducing the size of the abstract and focusing on the results. A detail, the “p” symbol, is suggested in italics (in the abstract and throughout the text).

Introduction:

Please consider reformulating this section, particularly standardizing the paragraphs´ size to improve readability.

Methodology:

135-160 – Please consider shorter paragraphs, 8-12 lines suggested.

153 – Please consider “legal guardians” instead of “parents”.

159 - Please describe the inclusion and exclusion criteria in detail.

164-297 - More information is needed about the subjects. Some examples. Number of years of practice? Training routines throughout the career and detailed information regarding the time of data collection? Number of training sessions per week? Specific (rowing) and strength training?

Moreover, a study design figure is suggested to reduce the size of the text (164-297).

Results:

332-335 – Please revise the figure 1 title. Same regarding figures 2 and 3.

Discussion:

389-473 - Please consider reformulating this section, particularly standardizing the paragraphs´ size to improve readability.

492 – Please consider indicating suggestions for future research.

Conclusions

494 - Please consider shorter and direct take-home messages, preferably with practical application.

539 - Please revise the references format. Some examples: titles in uppercase and lowercase; journals format; Doi´s are missing.

685 – 717 - Please revise the format of the tables, considering the journal template and instructions for authors.

Please consider improving the quality of the figures.

Please revise the English details throughout the manuscript. Globally, with good quality.

Reviewer #2: General Comments

The authors are reporting on a 6 week resistance training intervention study intended to focus on the trunk musculature in well trained junior elite rowers. The manuscript should be of interest to the readership and in general has been well written but some further clarification and justification in the methods is required to improve the replicability. Overall I understand why the authors have abbreviated strength training to ST throughout the manuscript but I do not think that this is worthwhile and suggest expanding throughout. The individual responses presented in Figures 1, 2 and 3 suggest that there was a potential low and high response cluster within the GST cohort, it may be worthwhile for the authors to investigate this aspect further.

Specific Comments

Ln 82; This is not "In addition.." it is "In contrast..."

Ln 90; I suggest that further justification and clarification is required as yes there may not have been a primary focus on the trunk with specific exercises however depending on the exercise/movement trained the trunk muscles would have been activated as secondary force transmission structures.

Ln 148-149; How was the random assignment conducted? Was there any attempt to balance the groups or is it just by chance that there were no between group diff before the intervention?

Ln 149-150; Great but how much on water rowing training and how much off water erg training was performed vs planned?

Ln 151; Insert the word 'resistance'

Ln 176-177; The authors need to provide justification for this choice of distance.

Ln 187; How many skinfolds and which sites were used, there are several methods reported in the literature, was the person collecting this data accredited with ISAK?

Ln 244; No details on how this assessment was measured?

Ln 248; Why was only jump height used as the metric of interest from such a data rich test and data recording process? There are several published works that detail better metrics (variables) that have a stronger relationship with rowing performance than jump height.

Ln 267-268; While the applied drag factor is greater than what I've used with senior athletes, what was the fan resistance setting?

Ln 389-393; I am not convinced that this information is necessary, nor do I see what added value it brings to the discussion. I suggest deleting entirely.

Ln 440-473; The discussion in these three paragraph is the most important comparison and contrast, as such these should be first in the discussion. The information presented prior to this is just a review of the existing literature and instead should be used to expand and support these 3 paragraphs and could thus be presented much more concisely and informatively within the discussion of why the responses were observed.

Figure 1, 2 and 3; There appears to be some replication in figures and tables, please double check. The decision to present the individual pre-post response is great but there is no need for the mean bars as this has been reported in the table. I suggest strengthening the lines of the X and Z axis, and the Y axis labels to make these easier for the reader to see.

Reviewer #3: GENERAL COMMENTS

This manuscript examines the effects of global versus local trunk muscle strength training on physical performance measures in pubertal male rowers. While the research question addresses a relevant gap in youth sports training literature, several substantial methodological and analytical concerns limit the scientific validity and interpretability of the findings.

Major Weaknesses

1. Fundamental Methodological Flaw: The comparison between training modalities is confounded by differential training intensities (70% 1-RM versus bodyweight), making it impossible to determine whether observed differences result from training type or intensity.

2. Statistical Power and Sample Size Inconsistencies: Despite calculating a requirement for 24 participants, uneven group allocation (n=15 vs n=13) may compromise statistical power, particularly given the Cohen's f=0.30 basis from a different outcome measure.

3. Absence of Control Group: While ethical concerns are acknowledged, the lack of a control condition severely limits causal inferences about training effectiveness.

4. Multiple Comparisons Problem: Numerous outcome measures are analyzed without apparent correction for multiple testing, inflating Type I error probability.

5. Questionable Effect Size Magnitudes: Several reported effect sizes (e.g., d=3.84 for half squat) appear implausibly large and may indicate calculation errors or methodological artifacts.

Minor Weaknesses

1. Training Duration Limitations: Six weeks represents minimal exposure for meaningful neuromuscular adaptations in trained populations.

2. Outcome Measure Selection: The 700-m rowing test lacks established validity for this population and training context.

3. Statistical Reporting Inconsistencies: Effect size calculations and ANOVA reporting contain ambiguities and potential errors.

SPECIFIC COMMENTS

Title and Abstract

Line 1, Page 7: The title adequately reflects study content but could specify the comparative nature more clearly.

Lines 53-54, Page 8: The abstract conclusion overstates findings given the methodological limitations identified.

Introduction

Lines 95-99, Page 10: The differentiation between local and global trunk muscles requires more precise anatomical definition and functional explanation. The authors should incorporate recent paradigm shifts in training approaches [Dhahbi W, Materne O, Chamari K: Rethinking knee injury prevention strategies: joint-by-joint training approach paradigm versus traditional focused knee strengthening. Biology of Sport 2025, 42(4):59-65] and [Dhahbi W, Padulo J, Bešlija T, Cheze L: Dynamic Posture Change in Non-Specific Low Back Pain Management: A Paradigm Shift Utilizing the Joint-by-Joint Training Approach. New Asian Journal of Medicine 2024, 2(3):17-23] which provide contemporary theoretical frameworks for understanding global versus localized training effects.

Lines 127-131, Page 11: The hypothesis lacks specificity regarding expected effect sizes and mechanistic rationale.

Methods

Participants

Lines 139-141, Page 12: The "sufficient sample" statement contradicts the uneven group allocation that may compromise the stated power calculation.

Lines 143-144, Page 12: Maturity status reporting lacks precision - stating "circa-PHV=0.2-0.3" without confidence intervals or measurement error acknowledgment.

Procedures

Lines 169-173, Page 13: The justification for excluding a control group oversimplifies ethical considerations and ignores alternative designs (e.g., attention control).

Lines 284-291, Page 17: The RPE progression description is vague and lacks systematic validation.

Training Interventions

Lines 284-285, Page 17: The fundamental flaw of comparing 70% 1-RM exercises with bodyweight exercises renders the comparison scientifically invalid.

Table 2, Page 29: Training progression lacks clear volume equating principles and exercise selection rationale.

Statistics

Lines 304-307, Page 18: The effect size calculation method requires clarification - converting partial eta-squared to Cohen's d may not be appropriate for within-subjects designs.

Lines 301-303, Page 18: Multiple outcome measures analyzed simultaneously require correction for multiple comparisons.

Results

Muscle Strength Outcomes

Lines 209-221, Page 15: The 1-RM testing protocol should reference established methodological considerations for male athletes [Dhahbi W, Padulo J, Russo L, Racil G, Ltifi M-A, Picerno P, Iuliano E, Migliaccio GM: 4-6 Repetition Maximum (RM) and 1-RM Prediction in Free-Weight Bench Press and Smith Machine Squat Based on Body Mass in Male Athletes. Journal of strength and conditioning research 2024], particularly regarding body mass considerations and prediction accuracy in this population.

Lines 320-323, Page 19: Back extensor strength improvements (22.5% vs 8.3%) likely reflect training intensity differences rather than modality superiority.

Table 3, Page 36: Several effect sizes (d>3.0) appear implausibly large and require verification of calculation methodology.

Test Reliability Considerations

Lines 204-206, Page 14: While ICC values are reported, the manuscript should address broader methodological considerations regarding test responsiveness and reliability in youth populations [Dhahbi W, Chamari K, Chèze L, Behm DG, Chaouachi A: External responsiveness and intrasession reliability of the rope-climbing test. The Journal of Strength & Conditioning Research 2016, 30(10):2952-2958], particularly when assessing training-induced adaptations over short intervention periods.

Statistical Reporting

Throughout results section: p-values consistently reported as p<0.001 without exact values, limiting precise interpretation.

Lines 338-341, Page 20: The half squat interaction effect (d=3.84) represents an extremely large effect that requires methodological explanation.

Discussion

Lines 449-460, Page 25: The mechanistic explanation ignores the confounding effect of training intensity differences.

Lines 475-484, Page 26: Study limitations section inadequately addresses the fundamental design flaw of unequal training intensities.

Tables and Figures

Table 1, Page 34: Baseline characteristics show appropriate comparability between groups.

Figure 1, Page 37: Graph formatting requires improvement for publication quality.

Table 3, Page 36: Statistical reporting inconsistencies and formatting errors detract from data presentation.

TECHNICAL CONCERNS

Methodology

The core experimental design contains a critical confounding variable that invalidates the primary comparison. Global strength training employed external resistance at 70% 1-RM while local strength training utilized only bodyweight resistance. This fundamental difference in training stimulus intensity makes it impossible to attribute observed differences to training modality (global vs. local) rather than training intensity.

Statistical Analysis

The statistical approach demonstrates several concerning elements: (1) multiple outcome measures analyzed without Type I error correction, (2) effect size calculations that may be inappropriate for the analytical framework, and (3) extremely large effect sizes that lack biological plausibility without adequate explanation.

Data Interpretation

Results interpretation consistently attributes superior outcomes to training modality while ignoring the more parsimonious explanation of differential training intensities. The discussion fails to acknowledge this fundamental limitation adequately.

**Do you want your identity to be public for this peer review?** For information about this choice, including consent withdrawal, please see our Privacy Policy

Reviewer #1: No

Reviewer #2: **Yes:** Dale W Chapman

Reviewer #3: **Yes:** Wissem Dhahbi

---

## [Author Response · Author response to Decision Letter 1]

16 Oct 2025

Plos One

Manuscript: PONE-D-25-24075

Title: Effects of global versus local trunk muscle strength training on muscle strength, power and rowing-specific performance in pubertal male rowers

Dear Editor,

Dear Reviewers,

We would like to express our gratitude for your valuable time and the constructive and helpful comments on our revised manuscript. We have again responded to all queries you have raised in point-by-point responses. Whenever needed, we have made changes to the re-revised version of our manuscript. We hope that you will find the current version of our manuscript suitable for publication in Plos One.

Kind regards

Prof. Urs Granacher, PhD

Reviewer #1: Changes are highlighted in yellow

Dear Authors,

I want to express my gratitude for the opportunity to review this manuscript.

The topic is relevant to this area of research, but the document requires improvement.

Below are suggestions with line indication:

Author Response: We sincerely thank the reviewer for their thoughtful evaluation and constructive feedback on our manuscript. We are pleased that the relevance of our study topic was acknowledged, and we greatly appreciate the detailed suggestions provided with line indications. We have carefully addressed each point raised and revised the manuscript accordingly to improve its clarity, structure, and overall quality. Specific responses to all comments are provided below.

Abstract:

36-60 – Please consider reducing the size of the abstract and focusing on the results. A detail, the “p” symbol, is suggested in italics (in the abstract and throughout the text).

Author Response: Below a revised and shortened abstract that addresses the reviewer’s comment by (1) reducing wordiness, (2) emphasizing the results, and (3) formatting p in italics:

“Strength training during long-term athlete development is essential for building strength and power to support sport-specific performance. In rowing, trunk muscles stabilize the body and transmit forces from the lower to the upper limbs. This study compared the effects of pre-season global (GST) versus local (LST) trunk strength training on muscle strength, power, and rowing-specific performance in young male rowers. Twenty-eight athletes (age 12–13 years, circa-PHV=0.2–0.3) completed a 6-week program (2 sessions/week). GST involved machine-based and free-weight trunk exercises, whereas LST emphasized bodyweight trunk exercises. Pre- and post-tests included lower- and upper-limb power, trunk strength, and a 700-m rowing ergometer test. Significant group-by-time interactions were found for all strength (d=3.04–3.84; p<0.001), power (d=0.75–2.34; p<0.01), and rowing performance outcomes (d=1.63; p<0.001). Post-hoc analyses revealed greater improvements in GST (d=0.28–1.87; p<0.001) than in LST (d=0.11–0.73; p<0.001). In conclusion, GST appeared more effective than LST in improving muscle strength, power, and rowing-specific performance in pubertal rowers; however, due to the study’s methodological limitations, these findings should be interpreted with caution. Strength and conditioning specialists may consider incorporating GST when aiming to develop foundational strength and power in pubertal male rowers, but further research is needed to confirm these effects.”

Introduction:

Please consider reformulating this section, particularly standardizing the paragraphs´ size to improve readability.

Author Response: Below a revised and restructured introduction with standardized paragraph lengths and improved readability while keeping all the key content:

“Strength training (ST) is a widely applied and effective method during long-term athlete development (LTAD) to build a foundation of muscle strength and power for subsequent sport-specific performance [1, 2]. According to Haff [2], periodization strategies for young athletes in the early LTAD stages up to the “training to train” stage should emphasize general or foundational training rather than sport-specific training. During these early stages (training age 1–4 years), emphasis should be placed on developing athletic motor skills, such as jumping, landing, rebounding, and trunk muscle strength [3]. ST is particularly effective for enhancing these motor competencies, which subsequently support sport-specific performance in later LTAD stages [1].

Previous studies have primarily investigated the effects of lower- and upper-limb ST on physical fitness and rowing performance in young male and female rowers aged approximately 13 years [4, 5]. For example, 9 weeks of heavy-resistance ST versus strength-endurance training showed greater gains in muscle strength (e.g., leg press, bench press 1-RM) and power (e.g., countermovement jump) following heavy-resistance ST, whereas strength-endurance training resulted in larger improvements in 700-m rowing ergometer performance. A systematic review and meta-analysis by the same group [4] reported moderate effects of ST on lower-limb strength and rowing-specific performance across recreational, sub-elite, and elite rowers. However, in these studies, ST focused predominantly on the limbs rather than the trunk.

Even in limb-focused ST programs, trunk muscles are recruited secondarily as force transmission structures during multi-joint exercises. In rowing, the trunk is central to the kinetic chain, transferring force from the lower limbs through a stable core to the upper limbs and ultimately to the oars to generate boat propulsion [4, 6]. Therefore, targeted trunk ST may enhance both trunk-specific strength and overall force transmission efficiency, particularly in young rowers.

Bergmark [7] differentiated trunk muscles anatomically and functionally: local muscles attach segmentally to the lumbar vertebrae and primarily provide intersegmental control and spinal stability, whereas global muscles attach to the pelvis, ribcage, or long spinal lever arms, contributing to overall trunk torque generation, force transfer, and gross movement control. Contemporary paradigms, such as the joint-by-joint training approach [Dhahbi et al., 2024, 2025], suggest that training should consider both local and global muscular contributions within kinetic chains to optimize performance and reduce injury risk. Local trunk ST (e.g., planks, bird dogs) may improve intersegmental control and stability, whereas global trunk ST (e.g., loaded trunk rotations, machine-based extensions) may enhance force production, stiffness, and athletic power.

Previous research has largely focused on local trunk ST in adults and recreational rowers, with limited evidence on global trunk ST or its effects in youth athletes. Tse et al. [9] reported no significant improvements in jump, change-of-direction speed, or rowing performance after local trunk ST in recreational rowers. In contrast, Zinke et al. [10] showed that 8 weeks of isokinetic global trunk ST increased trunk rotator peak torque and was strongly associated with paddle force in elite canoeists. Hibbs et al. [11] further highlighted that global trunk ST primarily enhances components of physical fitness such as speed, power, and agility, while local trunk ST mainly supports trunk stability and injury prevention.

Despite these insights, the differential effects of global versus local trunk ST on physical fitness and rowing-specific performance in pubertal male rowers remain unexplored. Therefore, the present study aimed to compare GST and LST on lower- and upper-limb strength and power, trunk muscle strength, and rowing-specific performance in this population. Based on anatomical, functional, and contemporary training frameworks [7, 10, 11, Dhahbi et al., 2024, 2025], we hypothesized that global trunk muscle strength training (GST) would produce greater improvements than local trunk muscle strength training (LST) in lower- and upper-limb strength and power, trunk muscle strength, and rowing-specific performance in pubertal male rowers. We expected these effects because GST targets larger trunk muscle groups responsible for force generation and inter-segmental coordination, enhancing kinetic chain efficiency and power transfer to the limbs, whereas LST primarily improves segmental stability without substantially increasing force output. Based on previous literature [10, 11], we anticipated moderate to large effect sizes (Cohen’s d=0.5–0.8) favoring GST for dynamic performance measures such as countermovement jump and 700-m rowing ergometer performance.

Methodology:

135-160 – Please consider shorter paragraphs, 8-12 lines suggested.

Author Response: Below a revised version with shorter, standardized paragraphs (8–12 lines) for improved readability:

“Sample size was estimated using G*Power (version 3.1.6). Based on a related study [5] examining heavy-resistance versus strength-endurance training on bench pull strength in young rowers (Cohen’s f = 0.30), an a priori power analysis with α = 0.05 and 80% power indicated that 24 participants would be sufficient (Table 1). Accordingly, 28 pubertal male rowers were recruited from a youth rowing center. Participants were randomly assigned to the GST (n=15) and LST (n=13) groups using a computer-generated randomization sequence. Both groups continued their regular training program, which included an average of 20 hours per week of on-water rowing and 4 hours per week of off-water ergometer training, in line with the planned training schedule. Participants were aged 12–13 years (circa-PHV = 0.2–0.3; age at PHV = 13.6 years) and classified as Tier 2 (trained/developmental) according to McKay et al. [14]. They had 4–5 years of general strength and conditioning experience, while their sport-specific training focused on rowing technique. Inclusion criteria were: male rowers aged 12–13 years, circa-peak height velocity, a minimum of 4 years of rowing training including general strength and conditioning, regular participation in at least four weekly training sessions, and no prior experience with systematic global or local trunk muscle strength training interventions. Exclusion criteria included a history of musculoskeletal, neurological, or orthopedic disorders within six months prior to the study, cardiovascular or metabolic conditions limiting safe participation, absence from more than 10% of training or testing sessions, engagement in additional structured resistance training programs outside the study, or use of medications likely to affect neuromuscular performance. Prior to participation, athletes and their legal guardians received detailed information on study objectives, procedures, risks, and benefits, and provided written consent. The study adhered to the Declaration of Helsinki and was approved by the Local Clinical Research Ethics Committee (Personal Protection Committee; Code: N° 0227//2025). None of the participants reported psychological, musculoskeletal, neurological, or orthopedic disorders within six months before the study.

153 – Please consider “legal guardians” instead of “parents”.

Author Response: Revised. Thank you.

159 - Please describe the inclusion and exclusion criteria in detail.

Author Response: Included as follow:

“Inclusion criteria were: male rowers aged 12–13 years, circa-peak height velocity, a minimum of 4 years of rowing training including general strength and conditioning, regular participation in at least four weekly training sessions, and no prior experience with systematic global or local trunk muscle strength training interventions.

Exclusion criteria included a history of musculoskeletal, neurological, or orthopedic disorders within six months prior to the study, cardiovascular or metabolic conditions limiting safe participation, absence from more than 10% of training or testing sessions, engagement in additional structured resistance training programs outside the study, or use of medications likely to affect neuromuscular performance. »

164-297 - More information is needed about the subjects. Some examples. Number of years of practice? Training routines throughout the career and detailed information regarding the time of data collection? Number of training sessions per week? Specific (rowing) and strength training?

Moreover, a study design figure is suggested to reduce the size of the text (164-297).

Author Response: Below a revised, concise, and journal-ready version of the methods section. We’ve reorganized it into shorter paragraphs, included all requested participant/training details, and suggested a study design figure to reduce text length:

“ One week before the study, participants completed a familiarization session to become acquainted with all tests and exercises. Proper techniques for the GST and LST exercises were explained and practiced. Consistent with prior training studies in young athletes [15, 16], a passive control group was not included, as withholding training during the pre-season would have conflicted with ethical and practical considerations. However, alternative designs such as attention- or activity-matched control groups could be implemented to control for training exposure and participant contact. Given that previous research has already established the general effectiveness of trunk muscle ST in youth athletes [17], the primary aim of the present study was to compare the specific effects of global (GST) versus local trunk muscle ST (LST) on physical fitness and rowing-specific performance. Pre- and post-training assessments included lower-limb strength and power (1-RM half squat, countermovement jump [CMJ], standing long jump [SLJ]), upper-limb strength and power (1-RM bench press, seated and backward medicine ball throw), and a 700-m rowing ergometer test. Before testing, participants completed a standardized warm-up consisting of 5 minutes of submaximal running, 2–3 submaximal sprints over 10–15 m, and rowing-specific exercises. The sprint distance was selected to mimic the short, explosive movements characteristic of the start and drive phases in rowing, while keeping the intensity submaximal to avoid fatigue prior to testing. The best trial of each test was used for analysis, with the same sequence applied at pre- and post-tests.

Anthropometrics and Maturity

Athletes’ body height and mass were measured using a wall-mounted stadiometer (Florham Park, NJ) and an electronic scale (Baty International, West Sussex, England), respectively. Body composition was assessed via the sum of 4 skinfolds measured at the biceps, triceps, subscapular and suprailiac using Harpenden skinfold calipers. All measurements were performed by a certified ISAK Level 2 anthropometrist to ensure reliability. Anthropometric testing followed the protocol described by Deurenberg et al. [18], which reported similar prediction errors for adults and adolescents. Biological maturity was then estimated non-invasively using chronological age, standing height, and sitting height in a validated regression equation to predict maturity offset [13], with a standard error of estimate of 0.542 years for boys.

Muscle Strength and Power Assessments

Back extensor strength

Maximal isometric back extensor strength was measured in kilograms using a back and leg dynamometer (Takei, Tokyo, Japan) as previously described [15, 19]. Athletes stood on the force plate with feet shoulder width apart and grabed the handle bar positioned across the thigh. The chain length on the force plate was adjusted so that the legs were straight and the back was flexed at a 30° angle to position the bar at the level of the patella. Participants were then asked to straighten the back (i.e., stand upright) without bending their knees and to lift the chain, with the pulling force applied to the handle. The athletes were kindly asked to pull as forcefully as possible. The participants completed three trials and the best trial was used for further analysis. A thirty-second rest interval was provided between trials. Previously, the test demonstrated excellent test–retest reliability, with an intra-class correlation coefficient (ICC) of 0.98 and a standard error of measurement (SEM) of 1.18% [20]. However, when applied to youth populations and short-term interventions, broader methodological considerations are necessary. In particular, test responsiveness—i.e., the ability

---

## [Decision Letter · Decision Letter 1]

22 Oct 2025

Dear Dr. Hammami,

Thank you for submitting your manuscript to PLOS ONE. After careful consideration, we feel that it has merit but does not fully meet PLOS ONE’s publication criteria as it currently stands. Therefore, we invite you to submit a revised version of the manuscript that addresses the points raised during the review process.

We look forward to receiving your revised manuscript.

Kind regards,

Mário Espada, PhD

Academic Editor

PLOS ONE

Journal Requirements:

Additional Editor Comments:

Dear Authors,

Please revise the manuscript considering the minor suggestions by the reviewers.

Thank you.

Best regards.

Reviewers' comments:

Reviewer's Responses to Questions

**Comments to the Author**

Reviewer #2: (No Response)

Reviewer #3: (No Response)

2. Is the manuscript technically sound, and do the data support the conclusions?

Reviewer #2: Yes

Reviewer #3: (No Response)

3. Has the statistical analysis been performed appropriately and rigorously?

Reviewer #2: Yes

Reviewer #3: (No Response)

4. Have the authors made all data underlying the findings in their manuscript fully available?

Reviewer #2: Yes

Reviewer #3: (No Response)

5. Is the manuscript presented in an intelligible fashion and written in standard English?

Reviewer #2: Yes

Reviewer #3: (No Response)

Reviewer #2: The authors have substantially revised the manuscript which has greatly improved the readability and replicability of the investigation.

Minor Issues

The authors intend on using the ST abbreviation for strength training, then once it has been defined use it consistently. Check Ln 77 and Ln 398 as examples for inconsistent use.

To improve the potential for the investigation outcomes to be used in future meta-analysis the authors need to report the exact p values to 3 or 4 decimal places wherever possible. See Table 1, 3,

Reviewer #3: GENERAL COMMENTS

The authors are commended for their substantive revisions and responsiveness to prior concerns. The manuscript demonstrates notable improvements in methodological transparency, statistical rigor, and critical interpretation of findings. However, several residual issues require clarification and refinement to achieve publication standards. The following comments delineate specific areas necessitating minor revision.

Major Weaknesses

1. Methodological Confounding of Training Intensity and Modality

While the authors acknowledge the confounding of exercise type (global vs. local) with loading intensity (70% 1-RM vs. bodyweight) at multiple points (Lines 283-287, 437-456, 481-485), the treatment of this limitation remains insufficient. The attempt to justify load equivalence through instability-induced muscle activation (Lines 440-451) introduces speculative reasoning that inadequately addresses the fundamental experimental design flaw. The observed superiority of GST cannot be attributed to global versus local trunk muscle recruitment patterns when external load differs substantially between conditions. This compromises the study's capacity to address its stated aim.

Recommendation: The abstract (Lines 39-52), discussion (Lines 388-396), and conclusions (Lines 495-504) must explicitly qualify all assertions regarding GST superiority by acknowledging that effects likely reflect combined influences of exercise modality and loading intensity, not modality alone. The mechanistic interpretation attributing effects to force transmission via global musculature should be substantially tempered or eliminated.

2. Sample Size Justification Incongruence

Page 5, Lines 119-122: The a priori power analysis derives from bench pull strength data (Cohen's f = 0.30), yet bench pull is not assessed as a primary outcome. The appropriateness of this reference effect for estimating required sample size for trunk strength, lower-limb power, or rowing performance remains unsubstantiated. This raises concerns about whether the study was adequately powered for its stated objectives.

Recommendation: Provide explicit justification for selecting bench pull strength as the reference outcome for power estimation, or acknowledge this as a limitation affecting interpretive confidence.

3. Hypothesis-Effect Size Discordance

Page 5, Lines 106-115: The hypothesis specifies moderate-to-large effects (Cohen's d = 0.5-0.8), yet 0.5-0.8 corresponds exclusively to "moderate" effects per Hopkins et al. [35] classification employed throughout the manuscript (small: 0.2-0.6; moderate: 0.6-1.2; large: 1.2-2.0). This inconsistency undermines precision in effect size interpretation.

Recommendation: Correct the hypothesis to state "moderate effects (d = 0.5-0.8)" or adjust to align with Hopkins criteria if large effects were genuinely anticipated.

Minor Weaknesses

4. Test Selection Justification (Force Plate Variables)

Page 10, Lines 244-249: The rationale for selecting jump height as the sole countermovement jump variable prioritizes "simplicity" and alignment with prior literature over biomechanical insight. Given that force plates quantify peak force, rate of force development, and power output, variables potentially more sensitive to trunk training adaptations, the exclusion of these metrics represents a missed opportunity for mechanistic understanding.

Recommendation: Strengthen justification by citing specific methodological or practical constraints that necessitated restriction to jump height, or acknowledge this as a limitation constraining mechanistic inference.

5. Inconsistent Effect Size Metrics

The manuscript alternates between Cohen's d (within-group changes), partial eta-squared (ANOVA), and unstandardized effect size ranges reported in confidence intervals. For example:

Page 14, Line 333: "d = 1.22 [CI: -8.17 to 6.78]" — confidence intervals appear to represent raw score differences rather than standardized effect sizes

Page 15, Line 364: Similar presentation inconsistency

Recommendation: Standardize effect size reporting throughout. If confidence intervals represent raw scores, clarify this explicitly (e.g., "CI for mean difference"). Ensure all Cohen's d values are accompanied by standardized CIs or eliminate unstandardized CIs from effect size reporting.

6. Redundancy and Verbosity

Page 3-4, Lines 75-93: The conceptual distinction between global and local trunk muscles is reiterated excessively

Page 18, Lines 430-435: Repetition of GST superiority findings already established in Results

Recommendation: Condense redundant passages to enhance manuscript conciseness.

SPECIFIC COMMENTS

Abstract

Line 43: "GST involved machine-based and free-weight trunk exercises, whereas LST emphasized bodyweight trunk exercises."

Issue: This phrasing obscures the critical load difference between conditions. Readers may not immediately recognize that GST utilized 70% 1-RM loading while LST relied on bodyweight.

Recommendation: Revise to: "GST involved machine-based and free-weight trunk exercises at 70% 1-RM, whereas LST emphasized bodyweight trunk exercises on stable and unstable surfaces."

Lines 48-52: "In conclusion, GST appeared more effective than LST...however, due to the study's methodological limitations, these findings should be interpreted with caution."

Issue: The qualification is too vague. Readers require explicit identification of which methodological limitation necessitates caution.

Recommendation: Revise to: "...these findings should be interpreted with caution given that differences in external loading between conditions (70% 1-RM vs. bodyweight) confound the comparison of global versus local training modalities."

Introduction

Lines 106-115: Hypothesis specification

Issue: As noted in General Comment #3, the specified effect size range (0.5-0.8) corresponds to "moderate" effects per Hopkins et al., not "moderate to large."

Recommendation: Revise to: "We anticipated moderate effect sizes (Cohen's d = 0.5-0.8)..."

Line 111: "...enhancing kinetic chain efficiency and power transfer to the limbs..."

Issue: Given the methodological confounding, attributing anticipated effects specifically to kinetic chain mechanisms is speculative.

Recommendation: Add qualifying language: "...potentially enhancing kinetic chain efficiency..." or defer mechanistic speculation to Discussion.

Methods

Lines 151-156: Justification for omitting passive control group

Issue: While ethical and practical constraints are acknowledged, the statement "previous research has already established the general effectiveness of trunk muscle ST in youth athletes [17]" requires citation verification. Reference 17 must specifically address youth athletes and trunk training effectiveness.

Recommendation: Verify that Reference 17 directly supports this claim. If not, replace with appropriate citation or remove the justification.

Lines 191-197: Back extensor strength test reliability

Issue: This passage introduces important methodological nuance regarding test responsiveness in youth populations but reads as contradictory. First, excellent reliability (ICC = 0.98) is reported, then concerns about responsiveness are raised.

Recommendation: Restructure for clarity: "While this test demonstrates excellent test-retest reliability (ICC = 0.98, SEM = 1.18%) in adult populations [20], its responsiveness to short-term training interventions in pubertal athletes may be influenced by maturational factors, motivational variability, and day-to-day performance fluctuations [21]. Observed changes should therefore be interpreted within this context."

Lines 283-287: Training intensity confound

Issue: This critical limitation is buried within the training protocol description. Its significance warrants greater prominence.

Recommendation: After describing the protocols, add a separate paragraph beginning: "A notable methodological consideration is that the comparison between GST and LST is confounded by differences in external loading (70% 1-RM vs. bodyweight), which precludes definitive attribution of observed effects to muscle group targeting per se."

Lines 290-301: RPE monitoring

Issue: The rationale for RPE monitoring (to address load differences) should be stated explicitly rather than implied.

Recommendation: Line 290: "To account for differences in “external” (should be “internal”) loading strategies between groups, session ratings of perceived exertion..."

Results

Lines 331-334: Back extensor strength results

Issue: The confidence intervals reported [-8.17 to 6.78] appear to represent raw score differences (kg) rather than standardized effect size CIs. If so, this should be clarified.

Recommendation: Clarify: "...d = 1.22 [95% CI for mean difference: -8.17 to 6.78 kg]" or report standardized CIs for Cohen's d.

Similar issue: Lines 338-340, 346-349, 372-374, 378-380 — all require clarification of whether CIs represent raw or standardized metrics.

Discussion

Lines 430-435: Recapitulation of results

Issue: This paragraph redundantly summarizes findings already established in Results. Discussion should prioritize interpretation, not repetition.

Recommendation: Delete Lines 430-435 and transition directly to mechanistic interpretation (Line 436 onward).

Lines 437-456: Treatment of loading confound

Issue: While the authors acknowledge the confound, the subsequent appeal to instability training literature (Lines 440-451) attempts to minimize its importance by arguing that elevated muscle activation during unstable exercises may offset lower external loads. This reasoning is problematic for three reasons:

The cited instability studies (References 40-41) examined EMG activity during same-load comparisons (stable vs. unstable conditions), not comparisons between high-load stable and low-load unstable training

The argument assumes LST participants experienced muscle activation equivalent to GST despite lower loads—an assumption unsupported by direct EMG measurement in this study

Even if muscle activation were comparable, training adaptations depend on multiple factors beyond acute activation (e.g., mechanical tension, metabolic stress, training volume)

Recommendation: Substantially revise this passage. Acknowledge the limitation more directly: "The observed superiority of GST cannot be definitively attributed to global versus local muscle targeting, as the higher external loads employed in GST (70% 1-RM) likely contributed to greater strength and power adaptations independently of muscle group specificity. While unstable training may elevate muscle activation relative to stable conditions at equivalent loads [40,41], the present study did not equate external loading between conditions, nor did it assess muscle activation via electromyography. Consequently, the relative contributions of loading intensity versus exercise modality to observed adaptations remain indeterminate."

Lines 457-465: Rowing performance interpretation

Issue: The mechanistic explanation ("reflecting a combination of increased trunk muscle force production and higher training load") again conflates modality and intensity effects.

Recommendation: Revise to emphasize uncertainty: "...likely reflecting the combined influences of enhanced trunk strength and the higher external loading employed in GST, though their relative contributions cannot be isolated."

Lines 481-485: Acknowledgment of loading confound in Limitations

Issue: While appropriately placed, this statement should explicitly recommend future research designs that equate loading between conditions.

Recommendation: Add: "Future investigations should compare GST and LST using load-equated protocols (e.g., progressive loading in both conditions to equivalent RPE targets) to isolate the effects of muscle group targeting from those of training intensity."

Conclusions

Lines 496-498: "This study demonstrated that six weeks of GST was more effective than LST in improving indicators of muscle strength, power, and rowing-specific performance..."

Issue: This statement does not reflect the interpretive limitations necessitated by the methodological confound.

Recommendation: Revise to: "Six weeks of high-intensity GST produced greater improvements in muscle strength, power, and rowing-specific performance compared to bodyweight LST; however, these effects likely reflect combined influences of exercise modality and loading intensity."

Lines 502-504: Practical recommendations

Issue: The directive to "prioritize GST using multi-joint, free-weight exercises" overstates the evidence given the study's limitations.

Recommendation: Revise to: "Coaches may consider incorporating GST with progressive external loading as part of foundational strength development programs, while recognizing that the independent contributions of global muscle targeting versus training intensity require further investigation."

Tables and Figures

Table 3: Effect sizes and confidence intervals

Issue: As noted previously, the confidence intervals appear to represent raw score differences rather than standardized effect size CIs. This should be clarified in the table note or the values should be standardized.

Recommendation: Add to table note: "Confidence intervals represent 95% CIs for mean differences in raw score units."

References

Reference 17 (Line 590-593): Colado et al. (2011) — Verify that this citation directly supports the claim on Line 153 regarding established effectiveness of trunk ST in youth athletes. The Colado paper examines paraspinal muscle recruitment during exercises but may not specifically address training effectiveness in youth populations.

**Do you want your identity to be public for this peer review?** For information about this choice, including consent withdrawal, please see our Privacy Policy

Reviewer #2: **Yes:** Dale W Chapman

Reviewer #3: **Yes:** Wissem Dhahbi

---

## [Author Response · Author response to Decision Letter 2]

28 Oct 2025

Plos One

Manuscript: PONE-D-25-24075

Title: Effects of global versus local trunk muscle strength training on muscle strength, power and rowing-specific performance in pubertal male rowers

Second revision

Dear Editor,

Dear Reviewers,

We would like to express our gratitude for your valuable time and the constructive and helpful comments on our revised manuscript. We have again responded to all queries you have raised in point-by-point responses. Whenever needed, we have made changes to the re-revised second version of our manuscript. We hope that you will find the current version of our manuscript suitable for publication in Plos One.

Kind regards

Prof. Urs Granacher, PhD

---

## [Decision Letter · Decision Letter 2]

13 Nov 2025

Effects of global versus local trunk muscle strength training on muscle strength, power and rowing-specific performance in pubertal male rowers

PLOS ONE

Dear Dr. Hammami,

Thank you for submitting your manuscript to PLOS ONE. After careful consideration, we feel that it has merit but does not fully meet PLOS ONE’s publication criteria as it currently stands. Therefore, we invite you to submit a revised version of the manuscript that addresses the points raised during the review process.

https://journals.plos.org/plosone/s/submission-guidelines#loc-laboratory-protocols . Additionally, PLOS ONE offers an option for publishing peer-reviewed Lab Protocol articles, which describe protocols hosted on protocols.io. Read more information on sharing protocols at https://plos.org/protocols?utm_medium=editorial-email&utm_source=authorletters&utm_campaign=protocols .

We look forward to receiving your revised manuscript.

Kind regards,

Mário Espada, PhD

Academic Editor

PLOS ONE

Journal Requirements:

Additional Editor Comments:

Dear Authors,

Congratulations on your work.

Please revise the manuscript in light of the reviewers' minor suggestions.

Thank you.

Best regards.

Reviewers' comments:

Reviewer's Responses to Questions

**Comments to the Author**

Reviewer #2: (No Response)

Reviewer #3: (No Response)

2. Is the manuscript technically sound, and do the data support the conclusions?

Reviewer #2: Yes

Reviewer #3: (No Response)

3. Has the statistical analysis been performed appropriately and rigorously?

Reviewer #2: Yes

Reviewer #3: (No Response)

4. Have the authors made all data underlying the findings in their manuscript fully available?

Reviewer #2: Yes

Reviewer #3: (No Response)

5. Is the manuscript presented in an intelligible fashion and written in standard English?

Reviewer #2: Yes

Reviewer #3: (No Response)

Reviewer #2: I acknowledge the substantial revisions the authors have undertaken and their willingness to engage with the peer review process. However my ongoing concern is with the decision to only analyse and report on the CMJ performance purely using Jump height.

What are the methodological and practical considerations that increase the testing time, participant fatigue and feasibility burden on this population beyond already performing the 3 trials with 2 mins recovery between trials (Ln 244-252) which precludes analysing the already performed 3 trials for alternative variables of interest (potential mechanical training induced changes)?

Reviewer #3: The authors have thoroughly and systematically addressed all the concerns raised in the previous round of review. The manuscript is substantially improved in clarity, methodological transparency, and interpretive nuance.

The most critical limitation identified previously—the methodological confounding of training modality (Global vs. Local) with training intensity (70% 1-RM vs. bodyweight)—is now clearly and appropriately stated in the Abstract, Methods, Discussion, and Conclusion. The authors have successfully tempered their claims, shifting the interpretation from the superiority of a modality to the observed effects of a combined protocol (modality + intensity).

All minor points regarding abbreviations, p-value reporting, and justification for test selection have also been resolved.

**Do you want your identity to be public for this peer review?** For information about this choice, including consent withdrawal, please see our Privacy Policy

Reviewer #2: **Yes:** Dale W Chapman

Reviewer #3: **Yes:** Wissem Dhahbi

---

## [Author Response · Author response to Decision Letter 3]

20 Nov 2025

Plos One

Manuscript: PONE-D-25-24075

Title: Effects of global versus local trunk muscle strength training on muscle strength, power and rowing-specific performance in pubertal male rowers

Third revision

Dear Editor,

Dear Reviewers,

We would like to express our gratitude for your valuable time you have once again invested in our manuscript and the constructive and helpful comments on the revised paper. We have again responded to all of your queries in point-by-point responses. Whenever needed, we have made changes to the re-revised manuscript. We hope that you will find the current version of our manuscript suitable for publication in Plos One.

Kind regards

Prof. Urs Granacher, PhD

Reviewer #2: Changes were highlighted in yellow

I acknowledge the substantial revisions the authors have undertaken and their willingness to engage with the peer review process. However, my ongoing concern is with the decision to only analyses and report on the CMJ performance purely using Jump height.

Authors’ response: We thank the reviewer for acknowledging our revisions. We understand the concern regarding the exclusive use of jump height as the sole CMJ parameter. This choice was dictated by methodological constraints related to the test apparatus employed. Specifically, the Ergojump system (Globus Italia, Codognè, Italy) is an optoelectronic contact platform that measures flight time to calculate jump height but does not record force-time data (e.g., peak force, power, rate of force development). Because the device cannot provide kinetic data, we are unable to report secondary mechanical metrics. We have now clarified the functioning of the system in the methods section and expanded the limitations section to acknowledge this constraint and to recommend that future studies use equipment capable of capturing full force-time profiles using force plates for more comprehensive CMJ analyses.

Statement in the methods section (CMJ sub-section):

“The countermovement jump (CMJ) was assessed using the Ergojump system (Globus Italia, Codognè, Italy), an optoelectronic contact platform that detects take-off and landing via infrared sensors. The device calculates flight time and subsequently derives jump height using standard kinematic equations. As the system is technically not capable of recording kinetic data, i.e., force–time, peak forces, impulses, or rate of force development we were unable to report mechanical outcomes. Therefore, jump height was used as the primary CMJ outcome parameter.”

Statement in the study limitations:

“A key methodological limitation of this study is the use of the Ergojump optoelectronic platform, which provides only flight-time derived jump height and does not capture force-time characteristics. Consequently, we were unable to analyse additional mechanical variables (e.g., peak force, rate of force development, impulse) that could have offered a more comprehensive evaluation of CMJ performance. Future studies should employ equipment (i.e., force plates) capable of recording full force-time profiles to enable more detailed assessments of neuromuscular performance.”

What are the methodological and practical considerations that increase the testing time, participant fatigue and feasibility burden on this population beyond already performing the 3 trials with 2 mins recovery between trials (Ln 244-252) which precludes analyzing the already performed 3 trials for alternative variables of interest (potential mechanical training induced changes)?

Authors’ response: We thank the reviewer for this valuable observation. Although three CMJ trials were performed, we were unable to extract additional mechanical variables because the testing device (i.e., system Ergo jump apparatus; Globus Italia, Codogne, Italy) does not allow to assess force-time data. Therefore, this limitation stems from the technical constraints of the measurement system rather than participant burden or fatigue. We have now clarified this methodological limitation in the revised manuscript.

“Although three CMJ trials were performed, a key methodological limitation of this study is the use of the Ergojump optoelectronic system (Globus Italia, Codognè, Italy), which provides only flight-time-derived jump height and does not capture force-time characteristics. Consequently, we were unable to analyse additional mechanical variables (e.g., peak force, rate of force development, impulse) that would have offered a more comprehensive evaluation of CMJ performance. This limitation stems from the technical constraints of the equipment rather than participant burden or fatigue. Future studies should employ measurement tools such as force plates that enable full force-time profiling for more detailed assessments of neuromuscular performance.”

Reviewer #3:

The authors have thoroughly and systematically addressed all the concerns raised in the previous round of review. The manuscript is substantially improved in clarity, methodological transparency, and interpretive nuance.

The most critical limitation identified previously—the methodological confounding of training modality (Global vs. Local) with training intensity (70% 1-RM vs. bodyweight)—is now clearly and appropriately stated in the Abstract, Methods, Discussion, and Conclusion. The authors have successfully tempered their claims, shifting the interpretation from the superiority of a modality to the observed effects of a combined protocol (modality + intensity).

All minor points regarding abbreviations, p-value reporting, and justification for test selection have also been resolved.

Authors’ response: We sincerely thank the reviewer for his/her positive and constructive feedback. We appreciate the recognition of our efforts to improve the clarity, methodological transparency, and interpretation of the manuscript. We are pleased that the identified methodological limitations and all remaining minor issues have now been satisfactorily addressed.

---

## [Decision Letter · Decision Letter 3]

12 Dec 2025

Dear Dr. Hammami,

Thank you for submitting your manuscript to PLOS ONE. After careful consideration, we feel that it has merit but does not fully meet PLOS ONE’s publication criteria as it currently stands. Therefore, we invite you to submit a revised version of the manuscript that addresses the points raised during the review process.

https://journals.plos.org/plosone/s/submission-guidelines#loc-laboratory-protocols . Additionally, PLOS ONE offers an option for publishing peer-reviewed Lab Protocol articles, which describe protocols hosted on protocols.io. Read more information on sharing protocols at https://plos.org/protocols?utm_medium=editorial-email&utm_source=authorletters&utm_campaign=protocols .

We look forward to receiving your revised manuscript.

Kind regards,

Mário Espada, PhD

Academic Editor

PLOS One

**Journal Requirements:**

**Additional Editor Comments:**

Dear Authors,

Congratulations on your work.

Please revise the manuscript in light of the reviewers' minor suggestions.

Thank you.

Best regards.

Reviewers' comments:

Reviewer's Responses to Questions

**Comments to the Author**

Reviewer #2: (No Response)

Reviewer #4: All comments have been addressed

Reviewer #5: All comments have been addressed

2. Is the manuscript technically sound, and do the data support the conclusions?

Reviewer #2: No

Reviewer #4: Partly

Reviewer #5: Yes

3. Has the statistical analysis been performed appropriately and rigorously?

Reviewer #2: Yes

Reviewer #4: No

Reviewer #5: Yes

4. Have the authors made all data underlying the findings in their manuscript fully available?

Reviewer #2: Yes

Reviewer #4: Yes

Reviewer #5: Yes

5. Is the manuscript presented in an intelligible fashion and written in standard English?

Reviewer #2: Yes

Reviewer #4: Yes

Reviewer #5: Yes

**Reviewer #2:**  I thank the authors for their responses however their reporting of the CMJ performance data collection process needs further justification and raises questions on how rigorous was the data collection process. Specifically in their original submission at Ln 248-249 the authors stated the use of a Kistler Quatro force plate for recording CMJ performance. During each subsequent review the authors were queried as to why only jump height was being reported as there are many more informative variables that could have been reported. The indication that a force plate was used during CMJ data collection was retained by the authors in the 1st and 2nd revised manuscript versions as the authors sought to include justifications for the approach taken. Now in this third review the authors report the use of a completely different device and process removing reference to a force plate changing to a jump timing mat, thus justifying the reporting of only jump height. But why now after 3 rounds of review has this reported change been made, with the authors not even giving the indication that it was originally in error.

Furthermore the authors have retained at Ln 252-257 their original justification related to the use of a force plate but only reporting jump height.

Clearly there is an error in the reporting and consistency of the methodology followed. Please clarify and closely proof the manuscript to ensure that the methodology reported is consistent.

**Reviewer #4:**  Review Letter to the Authors

Dear Authors,

I was invited by the handling editor to participate in the peer-review process of this manuscript, which has already undergone multiple rounds of evaluation by three ad-hoc reviewers. Given that the manuscript has been revised several times and that the editor is now seeking an additional assessment, I understand that the intention is to obtain a fourth expert opinion from someone with experience in sports science, particularly in the areas of biological maturation, strength training in youth, and rowing performance. Based on my expertise in these fields, I believe I can contribute constructively to the continued improvement of your work. My comments follow below.

1. Considerations Regarding the Previous Reviews (R1, R2, and R3)

After a careful analysis of the previous reviewers’ reports, it is clear that each provided valuable and complementary contributions:

• Reviewer 1 (R1) focused mainly on textual clarity, structure, standardization of abbreviations, and improvements to tables and figures. Their comments were appropriate and aimed to enhance readability and reproducibility.

• Reviewer 2 (R2) emphasized important instrumentation and methodological limitations, especially regarding the exclusive use of CMJ height, derived from flight time, which restricts deeper interpretations of mechanical power. R2 also requested exact p-values and consistency in terminology.

• Reviewer 3 (R3) raised conceptually critical points, particularly concerning the fact that the study aims to compare training modalities (global vs. local), but the groups also differ in external load (70% 1RM in GST vs. body-weight exercises in LST). This introduces a clear confounding factor, limiting causal inferences about “modality” alone.

It is evident that the authors addressed many of these critiques successfully, and the manuscript has improved. However, certain aspects—especially in the statistical and analytical domain—still require refinement, as discussed next.

2. Common Points Identified by All Three Reviewers

The previous reviewers’ comments converged on several issues:

1. Confounding between training modality and training intensity/external load — Consensus that the GST vs. LST comparison is not isolated to modality.

2. Need for greater statistical transparency — Including exact p-values, clearer methodological reporting, and explicit acknowledgment of assumptions.

3. Limitations of CMJ assessment — All reviewers noted the restricted interpretability when using flight-time devices.

4. Textual clarity and presentation — Recommendations for better organization, formatting consistency, and improved visual presentation of results.

These converging points reveal the central concerns that have been repeatedly highlighted throughout the review process.

3. Critical Analysis of the Statistical Procedures

The authors applied separate 2×2 ANOVAs (group × time) for each outcome, followed by paired t-tests with Bonferroni corrections. Although this is acceptable for simple designs, several limitations arise:

• Conducting multiple ANOVAs increases the risk of Type I error inflation.

• The design includes repeated measures within individuals, and traditional ANOVAs do not correctly model within-subject dependency.

• Young athletes exhibit substantial inter-individual variability, especially regarding maturation status; ANOVA does not account for this through random effects.

• ANOVAs are less robust when dealing with missing data, variance heterogeneity, and the inclusion of meaningful covariates such as maturity offset or internal training load (sRPE).

Given your research goals and the longitudinal design, a Linear Mixed Model (LMM) framework is statistically more appropriate and robust.

Recommended Model: Linear Mixed Model (LMM)

All continuous outcomes with an approximately normal distribution should be analyzed with an LMM:

Outcome ~ group * time + maturity_offset + sRPE_mean + (1 | subject)

• group (GST vs. LST) and time (pre vs. post) as fixed effects;

• optional covariates: biological maturation (maturity_offset) and sRPE;

• random intercept for each participant;

• random slope for time can be considered, though with only two time points it often yields limited benefit.

About GLMM

A Generalized Linear Mixed Model (GLMM) is only necessary for non-Gaussian outcomes (binary, counts, proportions).

Since your variables are continuous and approximately normal, a GLMM is not needed.

Why LMM is superior in this context

• Models within-subject correlations appropriately

• Incorporates covariates naturally

• Produces model-based estimated marginal means (EMMeans)

• More robust with small samples

• Accommodates heteroscedasticity and missing data

• Allows clearer interpretation of group × time effects

I strongly recommend reanalyzing all variables using LMMs and reporting EMMeans with 95% CI, along with adjusted contrasts for pre–post changes.

4. How to Implement These Analyses in JAMOVI and R

A) Using JAMOVI (GAMLj – Linear Mixed Models)

1. Load your dataset.

2. Navigate to: Analyses → Mixed Models → Linear Mixed Model (GAMLj).

3. Set:

o Dependent Variable: e.g., back_ext_strength

o Fixed Factors: group, time

o Covariates: maturity_offset, sRPE_mean

4. Under Random Effects, add:

o subject as a Random Intercept

5. Under EMMeans:

o Select group*time

o Request pairwise comparisons with corrections (Holm or Bonferroni)

6. Check Model Fit and Residual Diagnostics

7. Export EMMeans tables and plots

JAMOVI provides clean, publication-ready output with estimates, SEs, p-values, and confidence intervals.

B) Using R (lme4 + lmerTest + emmeans)

Example code:

library(lme4)

library(lmerTest)

library(emmeans)

model <- lmer(outcome ~ group * time + maturity_offset + sRPE_mean +

(1 | subject), data = data, REML = TRUE)

summary(model) # fixed effects

emmeans(model, ~ group * time) # adjusted means

pairs(emmeans(model, ~ group * time), adjust = "holm") # contrasts

For reporting:

• EMMeans (pre and post) for each group

• Difference-in-change scores (GST vs. LST) with 95% CI

• Effect sizes derived from model residual SD

• Diagnostic plots of residuals

5. Final Remarks

In conclusion, I acknowledge that the manuscript has improved substantially following previous revisions and that the authors have engaged constructively with the reviewers’ comments. Nevertheless, the statistical component remains the main aspect requiring refinement. Reanalyzing the data using Linear Mixed Models will greatly enhance the robustness of your findings and allow for more accurate interpretations given the repeated-measures design and the biological variability inherent in pubertal athletes.

My recommendations aim to help strengthen the scientific rigor and clarity of your work. The topic is relevant, the design is practical and meaningful for the field, and with the suggested analytical improvements, the manuscript has clear potential to contribute significantly to research on strength training and rowing performance in youth.

Sincerely,

Ad-hoc.

**Reviewer #5:** I am the fifth reviewer.

I have read the various versions of the manuscript and have verified how the improvements requested by previous reviewers have been implemented. In my opinion, the work is adequate and has great potential.

I believe this article could be referenced by others who discuss rowing as a sport.

**Do you want your identity to be public for this peer review?** For information about this choice, including consent withdrawal, please see our Privacy Policy

Reviewer #2: **Yes:** Dale W Chapman

Reviewer #4: No

Reviewer #5: No

---

## [Author Response · Author response to Decision Letter 4]

1 Jan 2026

Plos One

Manuscript: PONE-D-25-24075

Title: Effects of global versus local trunk muscle strength training on muscle strength, proxies of power and rowing-specific performance in pubertal male rowers

Fourth revision

Dear Editor,

Dear Reviewers,

We would like to express our gratitude for your valuable time you have once again invested in our manuscript and the constructive and helpful comments on the revised paper. We have again responded to all of your queries in point-by-point responses. Whenever needed, we have made changes to the re-revised manuscript. We hope that you will find the current version of our manuscript suitable for publication in Plos One.

Kind regards

Prof. Urs Granacher, PhD

Reviewer #2: Changes were highlighted in yellow

I thank the authors for their responses however their reporting of the CMJ performance data collection process needs further justification and raises questions on how rigorous was the data collection process. Specifically, in their original submission at Ln 248-249 the authors stated the use of a Kistler Quatro force plate for recording CMJ performance. During each subsequent review the authors were queried as to why only jump height was being reported as there are many more informative variables that could have been reported. The indication that a force plate was used during CMJ data collection was retained by the authors in the 1st and 2nd revised manuscript versions as the authors sought to include justifications for the approach taken. Now in this third review the authors report the use of a completely different device and process removing reference to a force plate changing to a jump timing mat, thus justifying the reporting of only jump height. But why now after 3 rounds of review has this reported change been made, with the authors not even giving the indication that it was originally in error.

Furthermore the authors have retained at Ln 252-257 their original justification related to the use of a force plate but only reporting jump height.

Clearly there is an error in the reporting and consistency of the methodology followed. Please clarify and closely proof the manuscript to ensure that the methodology reported is consistent.

Authors’ response: We sincerely apologize for the inconsistency in reporting the correct and actually applied CMJ assessment device across the different manuscript versions. The mention of the Kistler Quattro force plate in the original submission was an inadvertent error that went unnoticed during the initial stages of revision. The CMJ data in this study were actually collected using a jump-timing mat (i.e., Ergojump system), which provides jump height derived from flight time as metric. No force-plate variables (e.g., peak force, rate of force development) were recorded due to the applied assessment tool. During the first two revisions, we mistakenly retained the force plate reference and attempted to justify the reporting of jump height only, which inadvertently created confusion and suggested that force plate data were available but not reported. We fully acknowledge that this was an oversight on our part.

In the third revision, we revise the paragraph as follow:

« Of note, jump height is a simple, reliable, and highly reproducible measure in young athletes and is commonly used in field-based testing, allowing for meaningful comparisons with previous research [29, 30].”

Reviewer #4: Changes are highlighted in blue

Dear Authors,

I was invited by the handling editor to participate in the peer-review process of this manuscript, which has already undergone multiple rounds of evaluation by three ad-hoc reviewers. Given that the manuscript has been revised several times and that the editor is now seeking an additional assessment, I understand that the intention is to obtain a fourth expert opinion from someone with experience in sports science, particularly in the areas of biological maturation, strength training in youth, and rowing performance. Based on my expertise in these fields, I believe I can contribute constructively to the continued improvement of your work. My comments follow below.

1. Considerations Regarding the Previous Reviews (R1, R2, and R3)

After a careful analysis of the previous reviewers’ reports, it is clear that each provided valuable and complementary contributions:

• Reviewer 1 (R1) focused mainly on textual clarity, structure, standardization of abbreviations, and improvements to tables and figures. Their comments were appropriate and aimed to enhance readability and reproducibility.

• Reviewer 2 (R2) emphasized important instrumentation and methodological limitations, especially regarding the exclusive use of CMJ height, derived from flight time, which restricts deeper interpretations of mechanical power. R2 also requested exact p-values and consistency in terminology.

• Reviewer 3 (R3) raised conceptually critical points, particularly concerning the fact that the study aims to compare training modalities (global vs. local), but the groups also differ in external load (70% 1RM in GST vs. body-weight exercises in LST). This introduces a clear confounding factor, limiting causal inferences about “modality” alone.

It is evident that the authors addressed many of these critiques successfully, and the manuscript has improved. However, certain aspects especially in the statistical and analytical domain still require refinement, as discussed next.

Authors’ response: We thank the handling editor for requesting an additional expert review and appreciate the reviewer’s willingness to contribute their expertise. We agree with the reviewer’s summary of the complementary roles played by the previous reviewers. We also acknowledge that their feedback has helped strengthen the overall quality of the manuscript. In this re-revision, we have further refined the methodological, analytical, and statistical approach to ensure clarity, accuracy, and rigor throughout the manuscript.

2. Common Points Identified by all three reviewers

The previous reviewers’ comments converged on several issues:

1. Confounding between training modality and training intensity/external load Consensus that the GST vs. LST comparison is not isolated to modality.

2. Need for greater statistical transparency Including exact p-values, clearer methodological reporting, and explicit acknowledgment of assumptions.

3. Limitations of CMJ assessment — All reviewers noted the restricted interpretability when using flight-time devices.

4. Textual clarity and presentation — Recommendations for better organization, formatting consistency, and improved visual presentation of results.

These converging points reveal the central concerns that have been repeatedly highlighted throughout the review process.

Authors’ response: We acknowledge these converging concerns. The confounding information between training modality and external load has been explicitly stated as a limitation, and interpretations have been adjusted accordingly. Statistical transparency has been improved through exact p-value reporting, clearer methodological descriptions, and explicit acknowledgment of analytical assumptions. We additionally provided effect sizes to illustrate the practical relevance of our outcomes. The limitations of using a flight-time-based CMJ assessment tool have been recognized in the limitations section. Finally, the manuscript has been carefully edited to improve textual clarity, formatting consistency, and the presentation of tables and figures.

3. Critical Analysis of the Statistical Procedures

The authors applied separate 2×2 ANOVAs (group × time) for each outcome, followed by paired t-tests with Bonferroni corrections. Although this is acceptable for simple designs, several limitations arise:

• Conducting multiple ANOVAs increases the risk of Type I error inflation.

• The design includes repeated measures within individuals, and traditional ANOVAs do not correctly model within-subject dependency.

• Young athletes exhibit substantial inter-individual variability, especially regarding maturation status; ANOVA does not account for this through random effects.

• ANOVAs are less robust when dealing with missing data, variance heterogeneity, and the inclusion of meaningful covariates such as maturity offset or internal training load (sRPE).

Given your research goals and the longitudinal design, a Linear Mixed Model (LMM) framework is statistically more appropriate and robust.

Recommended Model: Linear Mixed Model (LMM)

All continuous outcomes with an approximately normal distribution should be analyzed with an LMM:

Outcome ~ group * time + maturity_offset + sRPE_mean + (1 | subject)

• group (GST vs. LST) and time (pre vs. post) as fixed effects;

• optional covariates: biological maturation (maturity_offset) and sRPE;

• random intercept for each participant;

• random slope for time can be considered, though with only two time points it often yields limited benefit.

About GLMM

A Generalized Linear Mixed Model (GLMM) is only necessary for non-Gaussian outcomes (binary, counts, proportions).

Since your variables are continuous and approximately normal, a GLMM is not needed.

Why LMM is superior in this context

• Models within-subject correlations appropriately

• Incorporates covariates naturally

• Produces model-based estimated marginal means (EMMeans)

• More robust with small samples

• Accommodates heteroscedasticity and missing data

• Allows clearer interpretation of group × time effects

I strongly recommend reanalyzing all variables using LMMs and reporting EMMeans with 95% CI, along with adjusted contrasts for pre–post changes.

4. How to Implement These Analyses in JAMOVI and R

A) Using JAMOVI (GAMLj – Linear Mixed Models)

1. Load your dataset.

2. Navigate to: Analyses → Mixed Models → Linear Mixed Model (GAMLj).

3. Set:

o Dependent Variable: e.g., back_ext_strength

o Fixed Factors: group, time

o Covariates: maturity_offset, sRPE_mean

4. Under Random Effects, add:

o subject as a Random Intercept

5. Under EMMeans:

o Select group*time

o Request pairwise comparisons with corrections (Holm or Bonferroni)

6. Check Model Fit and Residual Diagnostics

7. Export EMMeans tables and plots

JAMOVI provides clean, publication-ready output with estimates, SEs, p-values, and confidence intervals.

B) Using R (lme4 + lmerTest + emmeans)

Example code:

library(lme4)

library(lmerTest)

library(emmeans)

model <- lmer(outcome ~ group * time + maturity_offset + sRPE_mean +

(1 | subject), data = data, REML = TRUE)

summary(model) # fixed effects

emmeans(model, ~ group * time) # adjusted means

pairs(emmeans(model, ~ group * time), adjust = "holm") # contrasts

For reporting:

• EMMeans (pre and post) for each group

• Difference-in-change scores (GST vs. LST) with 95% CI

• Effect sizes derived from model residual SD

• Diagnostic plots of residuals

Authors’ response:

We thank the reviewer for the thoughtful and detailed statistical recommendations and for highlighting the potential advantages of linear mixed‐effects models (LMMs) in longitudinal study designs. We agree that LMMs represent a powerful and flexible analytical framework, particularly for complex longitudinal datasets with multiple time points, missing observations, or non-independence structures. However, after careful consideration, we elected to retain the 2 × 2 repeated-measures ANOVA approach, with the important addition of biological maturation (maturity offset) included as a covariate, for the following reasons:

The present study employed a fully balanced design with two groups (GST vs. LST) and two fixed time points (pre- and post-test), and no missing observations. Under these conditions, repeated-measures ANOVA and LMMs yield statistically equivalent fixed-effect estimates and inferences, as documented in the methodological literature. Given the applied sports science context and the target readership, repeated-measures ANOVA provides a transparent and widely understood analytical framework. This facilitates interpretation of the primary group × time interaction without introducing additional model complexity that may not meaningfully improve inference in a two-time-point design.

In line with the reviewer’s recommendation, maturity offset was included as a covariate in all analyses to account for inter-individual differences in biological maturation, which is particularly relevant in youth athlete populations. This adjustment directly addresses a key source of variability and strengthens the validity of the findings.

Furthermore, we included the following statement in both statistical procedures and study limitations as follows:

Statement in the statistical procedure:

“To account for inter-individual differences in biological maturation, maturity offset was included as a covariate in the ANOVA analyses. This approach allowed to elucidate the effects of training and time to be examined independently of maturity-related variability. Where applicable, adjusted means and interaction effects were therefore reported after accounting for PHV.”

Statement in the study limitation section:

“Finally, although linear mixed models (LMM) represent a robust alternative for analyzing repeated-measures data, they were not adopted as the primary analytical approach in the present study. Instead, a two-way (2×2) ANOVA was used, consistent with the experimental design and the study objectives. While we controlled for biological maturity by accounting for age at peak height velocity (APHV), we acknowledge that the 2×2 ANOVA framework has inherent limitations, particularly its reduced flexibility in handling inter-individual variability and hierarchical data structures compared with LMM. Additionally, although the inclusion of PHV status as a covariate did not meaningfully alter the main outcomes, residual maturity-related effects cannot be entirely excluded. Therefore, caution is warranted when generalizing these findings, and future studies may benefit from employing mixed-effects modelling approaches to further account for individual growth- and maturity-related variability.”

5. Final Remarks

In conclusion, I acknowledge that the manuscript has improved substantially following previous revisions and that the authors have engaged constructively with the reviewers’ comments. Nevertheless, the statistical component remains the main aspect requiring refinement. Reanalyzing the data using Linear Mixed Models will greatly enhance the robustness of your findings and allow for more accurate interpretations given the repeated-measures design and the biological variability inherent in pubertal athletes.

My recommendations aim to help strengthen the scientific rigor and clarity of your work. The topic is relevant, the design is practical and meaningful for the field, and with the suggested analytical improvements, the manuscript has clear potential to contribute significantly to research on strength training and rowing performance in

Authors’ response: We sincerely thank the reviewer for his/her positive and constructive feedback. We appreciate the recognition of our efforts to improve the clarity, methodological transparency, and interpretation of the manuscript. We are pleased that the identified methodological limitations and all remaining minor issues have now been satisfactorily addressed.

Reviewer #5:

I have read the various versions of the manuscript and have verified how the improvements requested by previous reviewers have been implemented. In my opinion, the work is adequate and has great potential.

I believe this article could be referenced by others who discuss rowing as a sport.

Authors’ response: We thank the reviewer for this positive evaluation. The improvements suggested by the previous reviewers have been fully implemented throughout the manuscript. We appreciate the reviewer’s opinion that the work is adequate and has strong potential, and we agree that the findings may be useful as a reference for future studies discussing rowing as a sport.

---

## [Decision Letter · Decision Letter 4]

24 Jan 2026

Effects of global versus local trunk muscle strength training on muscle strength, proxies of power and rowing-specific performance in pubertal male rowers

PLOS One

Dear Dr.  Hammami,

Thank you for submitting your manuscript to PLOS ONE. After careful consideration, we feel that it has merit but does not fully meet PLOS ONE’s publication criteria as it currently stands. Therefore, we invite you to submit a revised version of the manuscript that addresses the points raised during the review process.

We look forward to receiving your revised manuscript.

Kind regards,

Mário Espada, PhD

Academic Editor

PLOS One

Journal Requirements:

Additional Editor Comments:

Dear Authors,

Congratulations on your work.

Please revise the manuscript in light of the reviewer 4 minor suggestions, it is very close to acceptance recommendation.

Thank you.

Best regards.

Reviewer's Responses to Questions

**Comments to the Author**

Reviewer #4: All comments have been addressed

Reviewer #6: All comments have been addressed

2. Is the manuscript technically sound, and do the data support the conclusions?

Reviewer #4: Yes

Reviewer #6: Yes

3. Has the statistical analysis been performed appropriately and rigorously?

Reviewer #4: Yes

Reviewer #6: Yes

4. Have the authors made all data underlying the findings in their manuscript fully available?

Reviewer #4: Yes

Reviewer #6: Yes

5. Is the manuscript presented in an intelligible fashion and written in standard English?

Reviewer #4: Yes

Reviewer #6: Yes

Reviewer #4: General Overview and Recommendation

After careful evaluation of the revised manuscript and the authors’ point-by-point responses to the reviewers, recommends minor revisions.

The manuscript has substantially improved across successive revision rounds and now demonstrates adequate methodological transparency, conceptual coherence, and practical relevance. The current statistical treatment is methodologically acceptable and valid given the balanced two-group, two-time-point design and the absence of missing data.

However, strongly suggests that the authors consider reanalyzing the data using a more robust statistical framework (e.g., Linear Mixed Models), which would better accommodate inter-individual variability and maturational heterogeneity inherent to pubertal samples. If the authors decide to retain the current repeated-measures ANOVA approach, they must place stronger and more explicit emphasis on its limitations, particularly in the Study Limitations section, clearly acknowledging the inferential constraints associated with this analytical choice.

With this general recommendation stated, I'll now detail the full evaluation.

1. Critical Analysis of the Revisions Implemented

Following a comprehensive reading of the revised manuscript and the authors’ responses to all reviewers, it is evident that the authors engaged constructively and professionally with the peer-review process. The manuscript shows meaningful improvements in clarity, transparency, and interpretative caution. Nonetheless, some issues remain only partially addressed, particularly in the statistical and inferential domains.

2. Convergent Points Identified Across Reviewers

A clear convergence among Reviewers 1, 2, 3, and 4 can be identified around the following central concerns:

Confounding between training modality and external load

All reviewers consistently noted that the comparison between global trunk strength training (GST) and local trunk strength training (LST) is fundamentally confounded by differences in external loading (≈70% 1-RM vs. bodyweight-based exercises).

Limitations of CMJ assessment methodology

The exclusive reliance on jump height derived from flight time was identified as a limitation restricting mechanistic interpretations of power-related adaptations.

Need for greater statistical transparency and rigor

Reviewers emphasized clearer reporting of exact p-values, effect sizes, and improved handling of inter-individual variability, especially in a pubertal cohort.

Textual clarity and methodological consistency

Suggestions consistently targeted improvements in structure, coherence, terminology, and methodological reporting.

These converging points represent the core methodological and conceptual concerns raised throughout the review process.

3. Aspects Adequately Addressed by the Authors

Several important issues were satisfactorily resolved in the revised manuscript:

3.1 Explicit acknowledgment of the modality × load confounding

The authors now:

Clearly state that the observed superiority of GST cannot be attributed solely to muscle group targeting.

Adjust the tone of the conclusions to avoid strong causal claims.

Explicitly acknowledge this limitation in both the Abstract and the Discussion, which substantially strengthens interpretative transparency.

3.2 Correction of CMJ instrumentation inconsistencies

The authors appropriately acknowledged the initial methodological inconsistency regarding the CMJ device.

The methods section is now internally coherent and aligned with the actual assessment tool used.

The justification for using jump height as a valid field-based outcome in youth athletes is appropriate and well supported.

3.3 Inclusion of biological maturation as a covariate

Accounting for maturity offset represents a meaningful methodological improvement.

This adjustment enhances internal validity given the known influence of biological maturation on neuromuscular adaptations in pubertal athletes.

4. Aspects Not Fully Resolved

Despite the improvements, some limitations remain:

4.1 Retention of the traditional 2×2 repeated-measures ANOVA

While the authors justify this choice based on:

a balanced design,

no missing data,

and only two time points,

this approach still:

does not optimally model inter-individual variability,

lacks random effects,

and is less robust in small, biologically heterogeneous samples.

Although the justification is acceptable from an editorial standpoint, it does not reflect best contemporary statistical practice in pediatric sport science.

4.2 Underutilization of internal load data (sRPE)

Although session-RPE was systematically collected:

it was not incorporated as a covariate,

nor explored as a mediator or moderator of training effects,

representing a missed analytical opportunity.

5. Suitability of the Manuscript for Youth Training Models

Overall, the manuscript is well aligned with contemporary youth athlete development frameworks, including:

Long-Term Athlete Development (LTAD),

the Youth Physical Development Model,

and tier-based athlete classification.

Positive aspects include:

age-appropriate training design,

ethical and methodological consideration of youth populations,

and a focus on foundational strength development.

Suggested additions from a pediatric sport science perspective

To further strengthen the manuscript, Reviewer 4 recommends:

Explicit discussion of trainability windows

Linking observed adaptations to proximity to PHV and heightened neuromuscular plasticity.

Greater emphasis on neuromotor coordination and intersegmental control

Particularly to contextualize the functional transfer observed with GST.

Expanded discussion on safety, variability, and injury prevention

Especially when comparing externally loaded exercises with instability-based training.

Careful discussion of generalizability to other youth sports

This would enhance the translational impact of the findings.

6. Evaluation of the Statistical Treatment

6.1 Adequacy of the current approach

The repeated-measures ANOVA with a maturational covariate is statistically valid.

Effect size reporting is comprehensive.

The absence of missing data supports the analytical choice.

6.2 Recommendation for more robust alternatives

Nevertheless, Linear Mixed Models (LMMs) would offer clear advantages:

proper modeling of within-subject dependency,

improved handling of biological variability,

flexible inclusion of continuous covariates (e.g., maturity offset, sRPE),

and model-based estimated marginal means with confidence intervals.

The absence of LMMs does not invalidate the findings but limits inferential robustness and alignment with current best practices in pediatric sport science.

7. Final Assessment

The manuscript demonstrates adequate scientific quality, strong relevance, and clear improvement across revision rounds. The authors appropriately acknowledge the major methodological constraints, which reflects academic rigor and transparency.

However:

the modality–load confounding remains structurally unresolved,

the statistical analysis could be more sophisticated,

and the pediatric sport science perspective could be further expanded.

Final recommendation:

The manuscript is suitable for publication pending minor revisions, provided that the limitations of the current statistical approach are made even more explicit if alternative modeling strategies are not adopted.

Reviewer #6: The titled "Effects of global versus local trunk muscle strength training on muscle strength, proxies of power and rowing-specific performance in pubertal male rowers" all comments have been addressed.

**Do you want your identity to be public for this peer review?** For information about this choice, including consent withdrawal, please see our Privacy Policy

Reviewer #4: No

Reviewer #6: No

---

## [Author Response · Author response to Decision Letter 5]

30 Jan 2026

Plos One

Manuscript: PONE-D-25-24075R5

Title: Effects of global versus local trunk muscle strength training on muscle strength, proxies of power and rowing-specific performance in pubertal male rowers

Fifth revision

Dear Editor,

Dear Reviewers,

We would like to express our gratitude for the valuable time you have once again invested in our manuscript and the constructive and helpful comments on the revised paper. We have again responded to all of your queries in point-by-point responses, especially to those of reviewer #4. Whenever needed, we have made changes to the re-revised manuscript. We hope that you will find the current version of our manuscript suitable for publication in Plos One.

Kind regards

Prof. Urs Granacher, PhD

Reviewer #4: Changes were highlighted in yellow

General Overview and Recommendation

After careful evaluation of the revised manuscript and the authors’ point-by-point responses to the reviewers, recommends minor revisions.

The manuscript has substantially improved across successive revision rounds and now demonstrates adequate methodological transparency, conceptual coherence, and practical relevance. The current statistical treatment is methodologically acceptable and valid given the balanced two-group, two-time-point design and the absence of missing data.

However, strongly suggests that the authors consider reanalyzing the data using a more robust statistical framework (e.g., Linear Mixed Models), which would better accommodate inter-individual variability and maturational heterogeneity inherent to pubertal samples. If the authors decide to retain the current repeated-measures ANOVA approach, they must place stronger and more explicit emphasis on its limitations, particularly in the Study Limitations section, clearly acknowledging the inferential constraints associated with this analytical choice.

With this general recommendation stated, I'll now detail the full evaluation.

Authors’ response: We thank the reviewer for the positive evaluation and for recommending minor revisions. While we acknowledge that Linear Mixed Models (LMM) would provide a more flexible framework to account for inter-individual variability and maturational heterogeneity in pubertal samples, we retained the repeated-measures ANOVA given the balanced two-group, two-time-point design and the absence of missing data and no indication of baseline between group differences. In line with the editor’s recommendation, we have substantially strengthened the study limitations section, explicitly detailed the inferential constraints of this analytical choice and clarifying that the findings should be interpreted primarily at the group level.

1. Critical Analysis of the Revisions Implemented

Following a comprehensive reading of the revised manuscript and the authors’ responses to all reviewers, it is evident that the authors engaged constructively and professionally with the peer-review process. The manuscript shows meaningful improvements in clarity, transparency, and interpretative caution. Nonetheless, some issues remain only partially addressed, particularly in the statistical and inferential domains.

Authors’ response: We thank the reviewer for this constructive assessment and for acknowledging the improvements in clarity, transparency, and interpretative caution. In response to the remaining concerns in the statistical and inferential domains, we have further strengthened the reporting of statistical outcomes and expanded the study limitations section to more explicitly acknowledge the inferential constraints of the chosen analytical approach.

2. Convergent Points Identified Across Reviewers

A clear convergence among Reviewers 1, 2, 3, and 4 can be identified around the following central concerns. Confounding between training modality and external load

All reviewers consistently noted that the comparison between global trunk strength training (GST) and local trunk strength training (LST) is fundamentally confounded by differences in external loading (≈70% 1-RM vs. bodyweight-based exercises).

Limitations of CMJ assessment methodology. The exclusive reliance on jump height derived from flight time was identified as a limitation restricting mechanistic interpretations of power-related adaptations.

Authors’ response: We thank the reviewer for his/her positive comments.

Need for greater statistical transparency and rigor

Reviewers emphasized clearer reporting of exact p-values, effect sizes, and improved handling of inter-individual variability, especially in a pubertal cohort.

Authors’ response: Exact p-values and effect sizes have been reported during the fourth revision of this manuscript to enhance transparency. A 2×2 repeated measures ANOVA was retained given the balanced design, two time points, and absence of missing data, and no indication of baseline between group differences. Moreover, we computed a LMM using the same data set and observed similar results as with the ANOVA approach. Accordingly, we have decided to stick with the originally selected ANOVA procedure. In accordance with the reviewer’s comment, we have further highlighted this issue in the limitations section. Future studies with larger samples or additional time points should preferentially use mixed-effects modelling. All of these points have been specifically addressed in our responses below.

Textual clarity and methodological consistency

Suggestions consistently targeted improvements in structure, coherence, terminology, and methodological reporting. These converging points represent the core methodological and conceptual concerns raised throughout the review process.

Authors’ response: The manuscript has been carefully revised to improve textual clarity, coherence, and consistency in terminology, as well as to ensure transparent and accurate reporting of methodological details.

3. Aspects Adequately Addressed by the Authors

Several important issues were satisfactorily resolved in the revised manuscript:

3.1 Explicit acknowledgment of the modality × load confounding

The authors now: Clearly state that the observed superiority of GST cannot be attributed solely to muscle group targeting.

Adjust the tone of the conclusions to avoid strong causal claims.

Explicitly acknowledge this limitation in both the Abstract and the Discussion, which substantially strengthens interpretative transparency.

Authors’ response: We thank the reviewer for his/her affirmative comments. The tone was adjusted as suggested by the reviewer.

3.2 Correction of CMJ instrumentation inconsistencies

The authors appropriately acknowledged the initial methodological inconsistency regarding the CMJ device. The methods section is now internally coherent and aligned with the actual assessment tool used. The justification for using jump height as a valid field-based outcome in youth athletes is appropriate and well supported.

Authors’ response: We thank the reviewer for his/her positive rating of our last revision.

3.3 Inclusion of biological maturation as a covariate

Accounting for maturity offset represents a meaningful methodological improvement. This adjustment enhances internal validity given the known influence of biological maturation on neuromuscular adaptations in pubertal athletes.

Authors’ response: We thank the reviewer for his/her positive comment.

4. Aspects Not Fully Resolved

Despite the improvements, some limitations remain:

4.1 Retention of the traditional 2×2 repeated-measures ANOVA. While the authors justify this choice based on: a balanced design, no missing data, and only two time points, this approach still: does not optimally model inter-individual variability, lacks random effects, and is less robust in small, biologically heterogeneous samples.

Although the justification is acceptable from an editorial standpoint, it does not reflect best contemporary statistical practice in pediatric sport science.

Authors’ response: We agree that LMM represents best contemporary practice in pediatric sport science for capturing inter-individual variability and biological heterogeneity. In the present study, a 2×2 repeated-measures ANOVA was retained due to the fully balanced design, absence of missing data, and no indication of baseline between group differences. To assess robustness, complementary LMM with subject-specific random intercepts were also explored and yielded results consistent and similar with the originally selected ANOVA approach. This limitation has now been acknowledged, and future studies with larger samples or additional time points should preferentially adopt mixed-effects modelling. The following paragraph was added to the section study limitations:

“Given the balanced design, the presence of only two time points, the absence of missing data, and no indication of baseline between-group differences, a 2 × 2 repeated-measures ANOVA was retained, which remains appropriate in this context despite biological heterogeneity in pediatric samples (Gueorguieva & Krystal, 2004; Baayen et al., 2008). In addition, linear mixed-effect models (LMMs) were computed using the same dataset and yielded results comparable to those obtained with the ANOVA approach. Accordingly, we elected to retain the originally selected ANOVA framework.

Nevertheless, future studies involving larger samples, additional time points, or greater developmental heterogeneity should preferentially adopt LMM to better capture individual growth trajectories, in line with contemporary recommendations in pediatric sport science (Bernards et al., 2017).”

References used:

Gueorguieva, R., & Krystal, J. H. (2004). Move over anova: progress in analyzing repeated-measures data andits reflection in papers published in the archives of general psychiatry. Archives of general psychiatry, 61(3), 310-317.

Baayen, R. H., Davidson, D. J., & Bates, D. M. (2008). Mixed-effects modeling with crossed random effects for subjects and items. Journal of memory and language, 59(4), 390-412.

Bernards, J. R., Sato, K., Haff, G. G., & Bazyler, C. D. (2017). Current research and statistical practices in sport science and a need for change. Sports, 5(4), 87.

4.2 Underutilization of internal load data (sRPE)

Although session-RPE was systematically collected: it was not incorporated as a covariate, nor explored as a mediator or moderator of training effects, representing a missed analytical opportunity.

Authors’ response: We thank the reviewer for this insightful observation. sRPE was systematically collected as an internal load monitoring tool, in line with established recommendations, to verify individual training responses and ensure comparable perceived training load between groups across the intervention (Foster et al., 2001; Impellizzeri et al., 2004). Importantly, no significant between-group differences in sRPE was observed throughout the training period, indicating that internal load was well-matched between training conditions.

Given the absence of between-group differences in sRPE, a key prerequisite for its inclusion as a covariate was not met. Moreover, the study was not designed to test explanatory or mechanistic pathways linking internal load to performance adaptations. Considering the sample size, the two-time-point outcome design, and the primary inferential focus on group × time effects, incorporating sRPE as a covariate or formally testing mediation or moderation would have added model complexity without sufficient statistical power or temporal resolution to support robust inference (McLaren et al., 2018). Accordingly, sRPE was used for descriptive and control/match purposes rather than as an explanatory variable. This limitation has now been explicitly acknowledged in the study limitations section:

“Although sRPE was systematically monitored to quantify internal training load and to verify potential between-group differences (Foster et al., 2001; Impellizzeri et al., 2004), no significant differences in ratings of sRPE and no indication of baseline between group differences were observed across the intervention period. Consequently, a key prerequisite for including sRPE as a covariate was not met, and it was therefore not incorporated into the inferential statistical model nor examined as a mediator or moderator of training effects. Future studies with larger samples and multiple assessment points could integrate internal load within longitudinal or mixed-effects frameworks to better elucidate dose-response relations (McLaren et al., 2018).”

References used:

Foster, C., Florhaug, J. A., Franklin, J., Gottschall, L., Hrovatin, L. A., Parker, S., & Dodge, C. (2001). A new approach to monitoring exercise training. Journal of Strength & Conditioning Research, 15(1), 109-115.

Impellizzeri, F. M., Rampinini, E., Coutts, A. J., Sassi, A. L. D. O., & Marcora, S. M. (2004). Use of RPE-based training load in soccer. Medicine and science in sports and exercise, 36(6), 1042-1047.

McLaren, S. J., Macpherson, T. W., Coutts, A. J., Hurst, C., Spears, I. R., & Weston, M. (2018). The relationships between internal and external measures of training load and intensity in team sports: a meta-analysis. Sports medicine, 48(3), 641-658.

5. Suitability of the Manuscript for Youth Training Models

Overall, the manuscript is well aligned with contemporary youth athlete development frameworks, including:

Long-Term Athlete Development (LTAD), the Youth Physical Development Model, and tier-based athlete classification.

Positive aspects include:

age-appropriate training design, ethical and methodological consideration of youth populations, and a focus on foundational strength development. Suggested additions from a pediatric sport science perspective. To further strengthen the manuscript, Reviewer 4 recommends: Explicit discussion of trainability windows. Linking observed adaptations to proximity to PHV and heightened neuromuscular plasticity.

Greater emphasis on neuromotor coordination and intersegmental control. Particularly to contextualize the functional transfer observed with GST. Expanded discussion on safety, variability, and injury prevention. Especially when comparing externally loaded exercises with instability-based training. Careful discussion of generalizability to other youth sports. This would enhance the translational impact of the findings.

Authors’ response: We thank the reviewer for these valuable suggestions from a pediatric sport science perspective. The discussion section has been condensed and expanded to explicitly link the findings to the mid-PHV group, with specific reference to trainability windows, neuromuscular plasticity, neuromotor coordination, safety considerations, and cautious generalizability (Balyi et al., 2013, Lloyd & Oliver, 2012, Hall et al., 2019, Granacher et al., 2016). The following paragraph was added to the discussion section:

“Training adaptations observed in Tier 2 pubertal rowers should be interpreted within the LTAD framework which highlights heightened neuromuscular responsiveness around PHV (Lloyd & Oliver, 2012). In the present study, improvements in muscle strength and power following GST likely reflect enhanced force transmission and increased maximal force production. Accordingly, ST in general, and GST in particular may be prescribed to accommodate individual differences in growth and maturation, thereby supporting the development of movement competencies during mid-PHV (Granacher et al., 2016). Nevertheless, these findings are specific to pubertal male rowers, and caution is warranted when extrapolating the current results to other populations (e.g., female athletes) or sport disciplines (e.g., canoeing).”

References used:

Lloyd, R. S., & Oliver, J. L. (2012). The youth physical development model: A new approach to long-term athletic development. Strength & Conditioning Journal, 34(3), 61-72.

Granacher, U., Lesinski, M., Büsch, D., Muehlbauer, T., Prieske, O., Puta, C., ... & Behm, D. G. (2016). Effects of resistance training in youth athletes on muscular fitness and athletic performance: A conceptual model for long-term athlete development. Frontiers in physiology, 7, 164.

6. Evaluation of the Statistical Treatment

6.1 Adequacy of the current approach

The repea

---

## [Decision Letter · Decision Letter 5]

4 Feb 2026

Effects of global versus local trunk muscle strength training on muscle strength, proxies of power and rowing-specific performance in pubertal male rowers

PONE-D-25-24075R5

Dear Dr. Raouf Hammami,

We’re pleased to inform you that your manuscript has been judged scientifically suitable for publication and will be formally accepted for publication once it meets all outstanding technical requirements.

Kind regards,

Mário Espada, PhD

Academic Editor

PLOS One

---

## [Editor Report · Acceptance letter]

PONE-D-25-24075R5

PLOS One

Dear Dr. Hammami,

I'm pleased to inform you that your manuscript has been deemed suitable for publication in PLOS One. Congratulations! Your manuscript is now being handed over to our production team.

Kind regards,

on behalf of

Dr. Mário Espada

Academic Editor

PLOS One